# DBT is a metabolic switch for maintenance of proteostasis under proteasomal impairment

Ran-Der Hwang[1,2†], YuNing Lu[1,2†], Qing Tang[1,2], Goran Periz[1,2], Giho Park[1,2], Xiangning Li[1,2], Qiwang Xiang[1,2], Yang Liu[1,2], Tao Zhang[1,2], Jiou Wang[1,2*]

[1]Department of Biochemistry and Molecular Biology, Bloomberg School of Public Health, Baltimore, United States; [2]Department of Neuroscience, School of Medicine, Johns Hopkins University, Baltimore, United States

*For correspondence:
jiouw@jhu.edu

†These authors contributed equally to this work

Competing interest: The authors declare that no competing interests exist.

## Abstract

Proteotoxic stress impairs cellular homeostasis and underlies the pathogenesis of many neurodegenerative diseases, including amyotrophic lateral sclerosis (ALS). The proteasomal and autophagic degradation of proteins are two major pathways for protein quality control in the cell. Here, we report a genome-wide CRISPR screen uncovering a major regulator of cytotoxicity resulting from the inhibition of the proteasome. Dihydrolipoamide branched chain transacylase E2 (DBT) was found to be a robust suppressor, the loss of which protects against proteasome inhibition-associated cell death through promoting clearance of ubiquitinated proteins. Loss of DBT altered the metabolic and energetic status of the cell and resulted in activation of autophagy in an AMP-activated protein kinase (AMPK)-dependent mechanism in the presence of proteasomal inhibition. Loss of DBT protected against proteotoxicity induced by ALS-linked mutant TDP-43 in *Drosophila* and mammalian neurons. DBT is upregulated in the tissues of ALS patients. These results demonstrate that DBT is a master switch in the metabolic control of protein quality control with implications in neurodegenerative diseases.

## eLife assessment

This **important** study discovered DBT as a novel gene implicated in the resistance to MG132-mediated cytotoxicity and potentially also in the pathogenesis of ALS and FTD, two fatal neurodegenerative diseases. The authors provided **convincing** evidence to support a mechanism by which loss of DBT suppresses MG132-mediated toxicity via promoting autophagy. This work will be of interest to cell biologists and biochemists, especially in the FTD/ALS field.

## Introduction

The maintenance of protein homeostasis is essential for cell viability, but protein often misfolds during the life of the cell as a result of destabilizing mutations, environmental stress, or metabolic challenges. Protein misfolding is a common theme in human neurodegenerative disorders, among a growing list of 'conformational diseases,' which often manifest with pathologies of proteinaceous inclusions in the affected tissues (*Prusiner, 2012*). These neurodegenerative disorders include Alzheimer's (AD), Parkinson's (PD), Huntington's (HD) diseases, ALS, and frontotemporal dementia (FTD) (*Ross and Poirier, 2004*). For instance, protein inclusions containing TAR DNA-binding protein 43 (TDP-43) are a pathological hallmark of several neurodegenerative diseases, including the majority of ALS cases and significant subsets of FTD and AD cases (*Neumann et al., 2006*; *Cairns et al., 2007*; *Lagier-Tourenne et al., 2010*). The accumulation of misfolded proteins can overwhelm the protein quality

control systems in the cell and lead to functional impairment or cell death (*Tyedmers et al., 2010*). Therefore, proteotoxicity is considered a critical step in the pathogenesis of relevant neurodegenerative diseases. However, the mechanisms through which the cell organizes its protein quality control systems to alleviate proteotoxicity are not fully understood.

To guard against proteotoxicity, the cell has evolved elaborate machinery that can detect and degrade abnormal proteins to maintain the health of its proteome. The protein degradation machinery, including the ubiquitin-proteasome system (UPS) and autophagy (*Varshavsky, 2017*), play a critical role in removing the misfolded or aggregated proteins. The ubiquitin-proteasome system (UPS), a selective proteolytic system based on the tagging of substrate proteins with ubiquitin molecules, plays a pivotal role in protein quality control (*Glickman and Ciechanover, 2002*). Autophagy is another major pathway for protein quality control, which mediates bulk or selective protein degradation or removal of protein aggregates in the cell (*Yamamoto and Yue, 2014*). Impairment of both the UPS and autophagic systems has been extensively implicated in the pathogenesis of neurodegenerative diseases (*Ciechanover and Brundin, 2003*; *Wong and Cuervo, 2010*). Although proteotoxicity is known to be counteracted by the quality control systems including UPS and autophagy (*Pohl and Dikic, 2019*), how these systems are regulated in a coordinated manner to mount an effective defense against proteotoxic stress remains poorly understood.

Here, in search of critical regulators of proteotoxicity and protein quality control, we conducted an unbiased genome-wide CRISPR-Cas9 screen for suppressors of cytotoxicity as a result of proteasomal impairment and then identified DBT as a major regulator of protein quality control. We found that loss of DBT robustly enhanced the survival of cells under stress induced by proteasomal impairment. The mechanisms through which loss of DBT conferred resistance to the proteasomal inhibition was through an AMPK-dependent signaling that retained autophagic activities when the proteasomal activity was diminished. Loss of DBT also protected against proteotoxicity in neurodegenerative models. Abnormal upregulation of DBT in ALS patient tissues suggests that the DBT-dependent protein quality control pathways are dysregulated in the human disease conditions. These results have revealed a previously unknown mechanism of maintaining proteostasis that has implications for understanding the pathogenesis of neurodegenerative diseases.

## Results

### Genome-wide screening identifies DBT as a potent suppressor of proteasomal inhibition-induced cytotoxicity

To identify critical regulators of proteotoxicity as a result of proteasomal impairment, we designed a CRISPR-Cas9 screen to search for suppressors that confer resistance to toxicity induced by proteasomal inhibition in mammalian cells (*Figure 1A*). The screen was performed on human retinal pigment epithelium (RPE1) cells, a genomically stable diploid cell line, through infection with single guide RNAs (sgRNAs) from the human GeCKO (Genome-Scale CRISPR Knock-out) lentiviral pooled library (*Sanjana et al., 2014*). A potent and reversible proteasome inhibitor, MG132 (*Kisselev and Goldberg, 2001*), which is a peptide aldehyde and analog of proteasomal substrates, was applied to induce proteotoxicity. The sensitivity of RPE1 cells to proteasome inhibition was tested with MG132 treatment at a range of concentrations, with 2 µM of MG132 observed to cause lethality in most of the cells within 6 days (*Figure 1B*). The CRISPR screen was then performed with 2 µM of MG132 as the optimized concentration for selecting suppressors that would confer resistance to the proteasomal inhibition-induced toxicity. The RPE1 cells were infected with the GeCKO pooled lentiviral sgRNAs and cultured in the presence of 2 µM of MG132 for 6 days, and the surviving cells were allowed to recover without MG132 and grow into single cell-derived colonies. These colonies were then harvested and sequenced for further analysis.

The single-cell-derived colonies were expanded and analyzed for their resistance to the proteasomal inhibition-induced cell toxicity. The cell line that had the most potent resistance phenotype was found to harbor a specific sgRNA against the DBT gene. Sanger sequencing of the genomic locus targeted by the sgRNA revealed a homozygous mutation, where a 1 bp insertion in exon 2 resulted in a premature stop codon in exon 3 (*Figure 1—figure supplement 1A*). The loss of the C-terminal 77 amino acids from the 482 a.a. full-length protein suggests that this is a null mutant of DBT (*Figure 1—figure supplement 1A*). The phenotype of the DBT knockout (KO) cell line was verified through

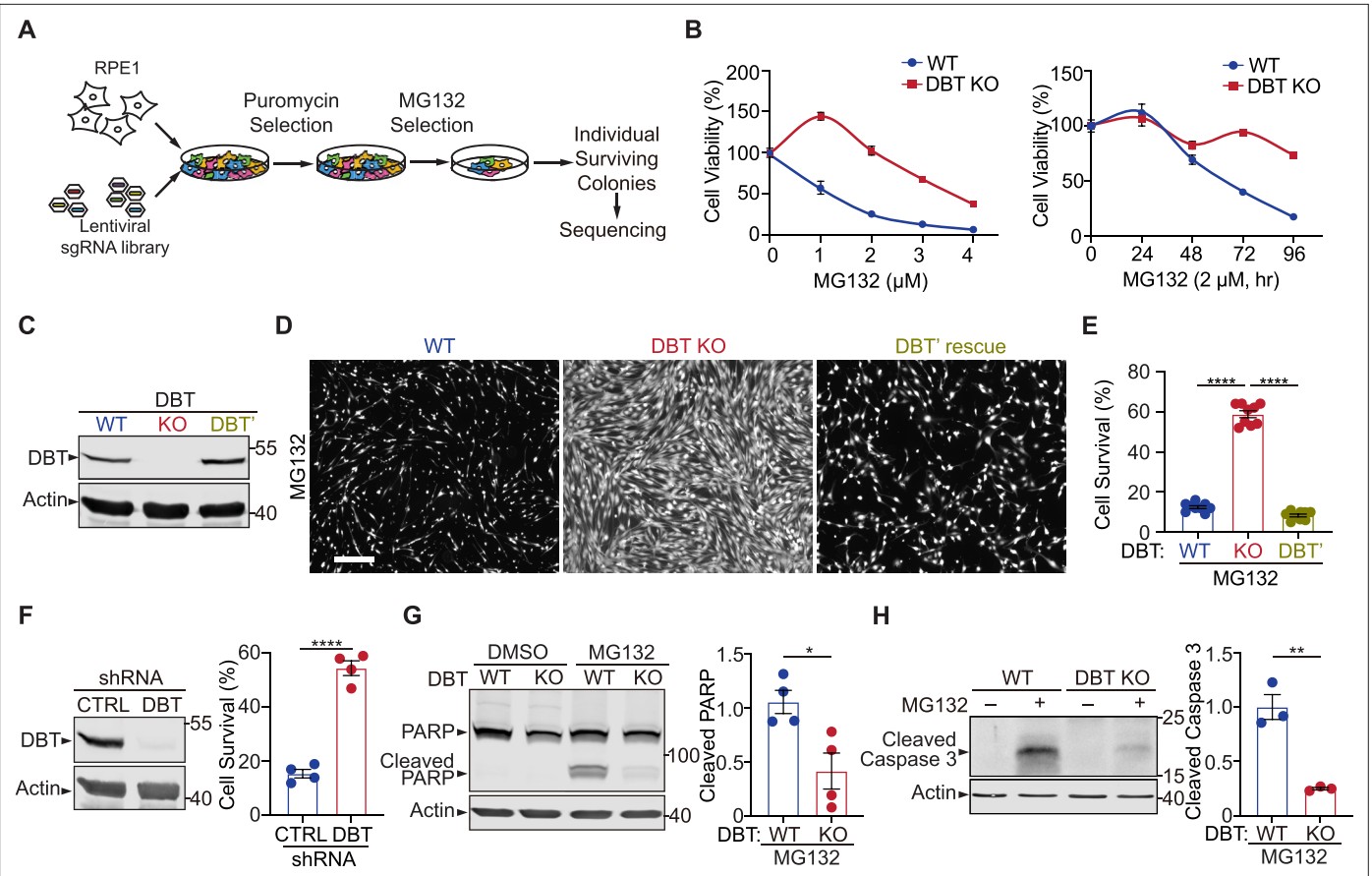

**Figure 1.** Genome-wide screen reveals that loss of dihydrolipoamide branched chain transacylase E2 (DBT) protects cells against the toxicity of proteasomal inhibition. (**A**) Workflow of the CRISPR screen in retinal pigment epithelium (RPE1) cells, which were transduced with a lentiviral Genome-Scale CRISPR Knock-out (GeCKO) single guide RNA (sgRNA) library and selected for the sgRNA expression and then survival after treatments with the proteasome inhibitor MG132. Individual surviving cell colonies were collected for sequencing and subsequent analysis. (**B**) Left: The cytotoxicity analysis of wild-type (WT) and DBT knockout (KO) RPE1 cells treated with MG132 at different doses for 96 hr (n=3). Right: The time course analysis of MG132-induced cytotoxicity in the WT and DBT KO cells (n=3). (**C**) Immunoblot analysis of WT RPE1, DBT KO, and DBT' cells. The DBT' cells expressed an engineered DBT cDNA that resisted DBT-targeted Cas9 cleavage and rescued the DBT expression in the KO cells. (**D**) Cell viability was measured by Calcein-AM staining in WT RPE1, DBT KO, and DBT' cells treated with MG132 (2 μM, 96 hr). Scale bar, 100 μm. (**E**) Quantification of the cell viability measured by Calcein-AM staining in (**D**) (n=9). (**F**) Left: Immunoblot analysis of RPE1 cells transfected with DBT shRNAs and non-targeting control shRNAs. Right: Quantification of the cell viability under treatment with MG132 (2 μM, 48 hr), as measured by Calcein-AM staining (n=4). (**G**) Immunoblotting and quantification of cleaved PARP as an MG132-induced cell death marker (n=4). (**H**) Immunoblotting and quantification of cleaved Caspase 3 as an MG132-induced cell death marker (n=3). Error bars represent means ± SEM. *p≤0.05; **p≤0.01; ****p≤0.0001.

The online version of this article includes the following source data and figure supplement(s) for figure 1:

**Source data 1.** Original and uncropped blots for *Figure 1C*.

**Source data 2.** Original and uncropped blots for *Figure 1F*.

**Source data 3.** Original and uncropped blots for *Figure 1G*.

**Source data 4.** Original and uncropped blots for *Figure 1H*.

**Figure supplement 1.** Schematics of CRISPR editing, dihydrolipoamide branched chain transacylase E2 (DBT) cDNA, and cell viability analysis.

a series of cell survival experiments, in which the cells were subjected to MG132 treatment at a range of concentrations or over a period of time, and the DBT KO cells exhibited the robust phenotype of resisting MG132-induced cytotoxicity when compared to the wild-type (WT) control cells (*Figure 1B*). Additionally, the resistance of DBT KO cells to proteasome inhibition-induced cytotoxicity was confirmed using another proteasome inhibitor, bortezomib. The results from the bortezomib assay were consistent with those of MG132, as the DBT KO cells consistently demonstrated better survival rates than the WT control cells (*Figure 1—figure supplement 1C and D*). To further confirm

the role of DBT in the cell survival phenotype through an independent approach of removing DBT protein, we knocked down DBT in RPE1 cells by using a small hairpin RNA (shRNA) against the gene and observed a similar phenotype, indicating that DBT deficiency indeed underlies the resistance of the cells to MG132-induced cell toxicity (*Figure 1F*).

Next, we asked whether the resistance of the DBT KO cells to MG132-induced cell toxicity could be rescued by exogenous DBT. We engineered a DBT cDNA (DBT') harboring seven synonymous point mutations in the gRNA-targeted region, which rendered the cDNA resistant to the Cas9 cleavage in the DBT KO cell line (*Figure 1C* and *Figure 1—figure supplement 1B*). The expression of DBT' in the DBT KO cells completely restored the cell lethality phenotype induced by MG132 treatment, as measured by the fluorescence signals from Calcein AM (*Figure 1D and E*), a cell-permeant dye that stains live cells. Moreover, the MG132-induced cell death can be measured by the cleavage of marker proteins such as PARP1 and Caspase 3 through immunoblotting analysis. While WT RPE1 cells exhibited pronounced cell death as indicated by an increase in cleaved PARP1 and Caspase 3 after treatment with 2 µM MG132 for 48 hr, loss of DBT markedly suppressed the cleavage of these cell death marker proteins (*Figure 1G and H*). Together, these results demonstrate that loss of DBT robustly suppresses cell toxicity induced by the proteasome inhibitor MG132.

## Loss of DBT prevents the accumulation of ubiquitinated proteins

Next, we sought to identify the underlying mechanism through which DBT regulates MG132-induced cytotoxicity. The primary action of MG132 is to block the proteolytic activity of the 26 S proteasome complex, which is responsible for the degradation of most ubiquitinated proteins. To assess the effect of MG132-induced proteasomal inhibition on protein ubiquitination in WT and DBT KO RPE1 cells, we performed immunoblotting analysis of ubiquitinated proteins, which showed an accumulation of poly-ubiquitinated proteins in the WT cells upon the MG132 treatment, whereas the MG132-induced changes in the levels of poly-ubiquitinated proteins in the DBT KO cells were significantly reduced (*Figure 2A*). Moreover, the relative reduction in the accumulation of poly-ubiquitinated proteins in the DBT KO cells compared with those in the WT cells was observed over a period of 3 days of the MG132 treatment (*Figure 2—figure supplement 1A*), when more than half of the WT cells had died, Furthermore, we performed immunofluorescence staining analysis on these cells using a ubiquitin-specific antibody. Although the ubiquitin signal was very low without MG132 treatment, there was a substantial accumulation of ubiquitin-positive proteins in the WT cells upon treatment with 2 µM of MG132 for 48 hr. However, the MG132-induced increase in ubiquitin immunofluorescence signals was greatly diminished in the DBT KO cells when compared with the WT cells (*Figure 2B*), suggesting that loss of DBT suppressed the MG132-induced cell death by preventing the accumulation of ubiquitinated proteins.

Ubiquitin modification is a universal feature of pathological protein aggregates observed in neurodegenerative diseases (*Alves-Rodrigues et al., 1998*). To measure the levels of aggregated proteins in WT or DBT KO RPE1 cells, we employed a fluorescent dye, ProteoStat, that detects protein aggregates by selectively intercalating into the cross-beta spine of quaternary structures typically found in misfolded and aggregated proteins (*Lesire et al., 2020*). The fluorescence staining indicated that the MG132 treatment triggered a substantial accumulation of aggregated proteins in the WT cells; however, the levels of the protein aggregates were significantly reduced in the DBT KO cells under the MG132 treatment (*Figure 2C*), consistent with the changes in the levels of poly-ubiquitinated proteins. These results demonstrate that loss of DBT prevents the accumulation of protein aggregates as a result of MG132-induced proteasomal inhibition.

## Loss of DBT preserves autophagy under proteasomal inhibition

Next, we asked whether loss of DBT suppressed the MG132-induced cell toxicity by enhancing the proteasomal activity. Using a luminogenic Proteasome-Glo substrate, we measured the chymotrypsin-like proteolytic activity of the proteasome in WT and DBT KO RPE1 cells. The DBT KO cells had similar proteasomal activity to that of WT cells under resting states. Notably, the MG132 treatment decreased the proteasomal activity in both WT and DBT KO cells but to the same degree (*Figure 2—figure supplement 1B*). Furthermore, the protein levels of proteasomal core subunits, such as PSMD1, PSMD5, and PSMD11, were unchanged between the WT and DBT KO cells before or after the MG132 treatment (*Figure 2—figure supplement 1C–F*). These results show that loss of DBT does

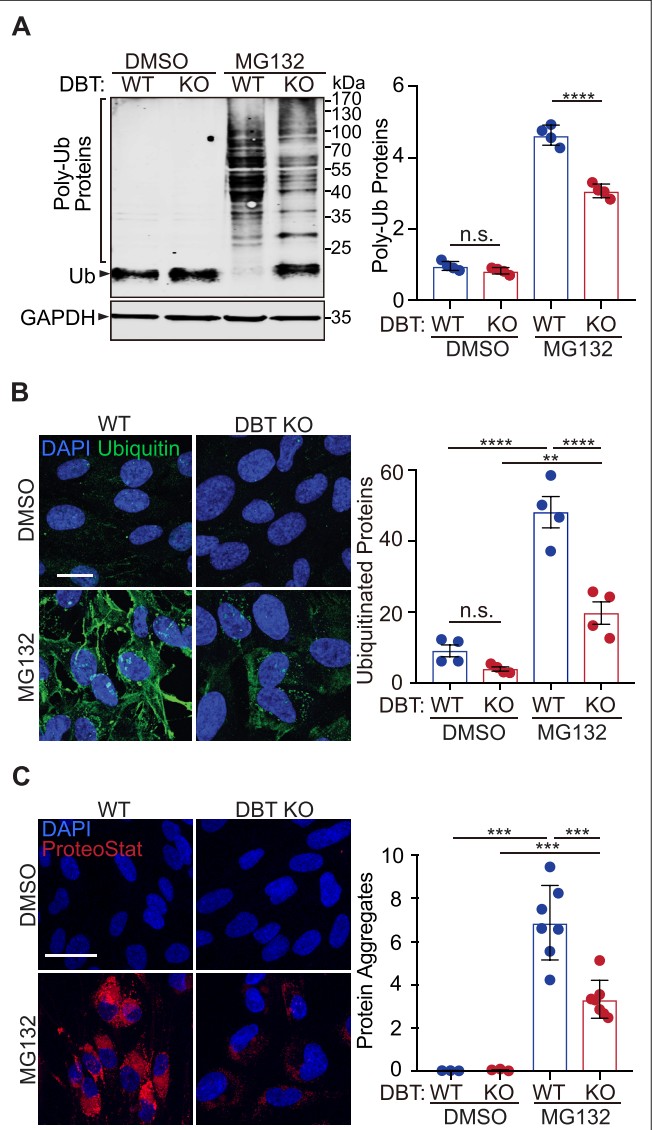

**Figure 2.** Loss of dihydrolipoamide branched chain transacylase E2 (DBT) decreased the accumulation of ubiquitinated proteins upon proteasomal inhibition. (**A**) Wild-type (WT) and DBT knockout (KO) retinal pigment epithelium (RPE1) cells treated with MG132 (2 μM, 48 hr) or the DMSO solvent control were analyzed for accumulation of ubiquitinated proteins upon proteasomal inhibition with denaturing SDS-PAGE. The bar graph represents quantification of the high-molecular-weight poly-ubiquitinated proteins (n=4). (**B**) The cells treated with MG132 (2 μM, 72 hr) or the DMSO solvent control were analyzed for the levels of ubiquitinated protein with immunostaining. The bar graph represents the quantification of the anti-ubiquitin immunofluorescent signals (n=4 independent groups, each consisting of seven cells). Scale bar, 10 μm. (**C**) The cells treated with MG132 (2 μM, 72 hr) or the DMSO solvent control were stained with a dye that detects protein aggregates. The bar graph represents the quantification of the ProteoStat signals (n=7 biological replicates of DBT KO cells and three replicates of WT control cells). Scale bar, 20 μm. Error bars represent means ± SEM. 'n.s.', no significance; **p≤0.01; ***p≤0.001; ****p≤0.0001.

The online version of this article includes the following source data and figure supplement(s) for figure 2:

**Source data 1.** Original and uncropped blots for *Figure 2A*.

**Figure supplement 1.** The enhanced clearance of poly-ubiquitinated proteins upon loss of dihydrolipoamide branched chain transacylase E2 (DBT) is not mediated by proteasomal degradation.

**Figure supplement 1—source data 1.** Original and uncropped blots for *Figure 2—figure supplement 1A*.

**Figure supplement 1—source data 2.** Original and uncropped blots for *Figure 2—figure supplement 1C*.

not promote proteasomal activity but prevents the accumulation of ubiquitinated proteins through a proteasome-independent mechanism.

The autophagy-lysosome pathway is an alternative mechanism for the degradation of ubiquitinated proteins (*Kraft et al., 2010*). Thus, we examined the status of autophagy in the WT and DBT KO RPE1 cells by measuring the levels of LC3II, a lipidated form of LC3 protein that serves as a maker for the autophagosome. In the absence of MG132 treatment, the LC3II levels were similar between WT and DBT KO cells. Treatment with the lysosome inhibitor Bafilomycin A1 blocks the degradation of LC3II, and the Bafilomycin A1-induced increase in LC3II was comparable between WT and DBT KO cells, indicating that the DBT KO cells had normal autophagy flux levels similar to those of WT cells under the resting condition (*Figure 3A–C*; *Loos et al., 2014*). Upon the treatment with 2 µM of MG132 for 48 hr, there was a significant increase in LC3II levels in WT cells; however, the Bafilomycin A1-induced inhibition of lysosomal degradation did not further increase the LC3II levels in the WT cells, indicating that the autophagy flux was severely impaired, thereby leading to the accumulation of LC3II in these cells (*Figure 3A–C*). By contrast, the DBT KO cells exhibited lower levels of LC3II than the WT cells upon the MG132 treatment, and unlike the WT cells, these DBT KO cells showed further increase in LC3II levels after the Bafilomycin A1-induced inhibition of lysosomal degradation, indicating that the DBT KO cells had a functioning autophagy flux under the MG132-treated condition (*Figure 3A–C*). Moreover, using another compound, vinblastine, that blocks the fusion between the autophagosome and the lysosome, we observed effects similar to those of Bafilomycin A1 that DBT KO cells showed further increase in the LC3II levels after the inhibition of lysosomal degradation (*Figure 3—figure supplement 1A*). Time course analyses of LC3II levels indicated that MG132-induced increases of LC3II could be observed as early as 4 hr after the treatment and that DBT KO cells consistently showed higher LC3II levels than the WT control cells (*Figure 3—figure supplement 1B C*). When DBT expression was reintroduced into the DBT KO cells, the level of LC3II was decreased (*Figure 3—figure supplement 1D*). These results suggest that loss of DBT enhances the function of the autophagy-lysosome pathway under the stress induced by proteasomal inhibition.

P62 is the autophagy receptor for ubiquitinated proteins (*Katsuragi et al., 2015*). Since p62 is degraded by the autophagy-lysosome pathway, its protein level is typically enhanced when the function of the autophagy-lysosome pathway is impaired. In the absence of MG132 treatment, the p62 levels were comparable between WT and DBT KO cells and were increased to similar levels upon the treatment with the lysosome inhibitor Bafilomycin A1, confirming that the DBT KO cells had normal autophagy functions under the resting condition (*Figure 3D–F*). However, under the MG132-induced proteotoxic condition, the WT cells showed a heightened level of p62 and there was no further increase in p62 after the Bafilomycin A1-induced inhibition of lysosomal degradation (*Figure 3D–F*). By contrast, with the MG132 treatment, the DBT KO cells exhibited lower levels of p62 than the WT cells and showed a further increase in p62 after the lysosomal inhibition, confirming that the DBT KO cells had preserved the autophagic functions under the MG132-treated condition (*Figure 3D–F*).

To further test whether the resistance of the DBT KO cells to the MG132-induced cytotoxicity was a result of the preserved autophagic activities, we treated the WT and DBT KO cells with Bafilomycin A1 and analyzed the cell survival. As mentioned above, loss of DBT protected cells from MG132-induced cell death; however, the inhibition of the autophagy-lysosome pathway with Bafilomycin A1 completely abolished the enhanced cell survival associated with loss of DBT (*Figure 3G*), indicating that the autophagy-lysosome pathway is required for the DBT-dependent regulation of proteotoxicity under proteasomal inhibition.

## DBT is a metabolic switch regulating proteotoxicity-dependent activation of AMPK

DBT is a core component of the branched-chain α-keto acid dehydrogenase (BCKD) enzyme complex, which functions in inner mitochondria to break down the essential branched-chain amino acids (BCAAs), leucine, isoleucine, and valine (*Brosnan and Brosnan, 2006*). The BCKD-mediated reaction is the rate-limiting and irreversible step in the catabolism of BCAAs into acetyl-CoA or succinyl-CoA, which can be used for energy production by the citric acid cycle (*Holeček, 2018*). To understand the mechanism through which DBT regulates MG132-induced cytotoxicity, we examined the metabolic and energetic status of WT and DBT KO RPE1 cells under the proteotoxic stress. First, BCAA levels were analyzed in the WT and DBT KO cells using a colorimetric assay. The DBT KO cells exhibited

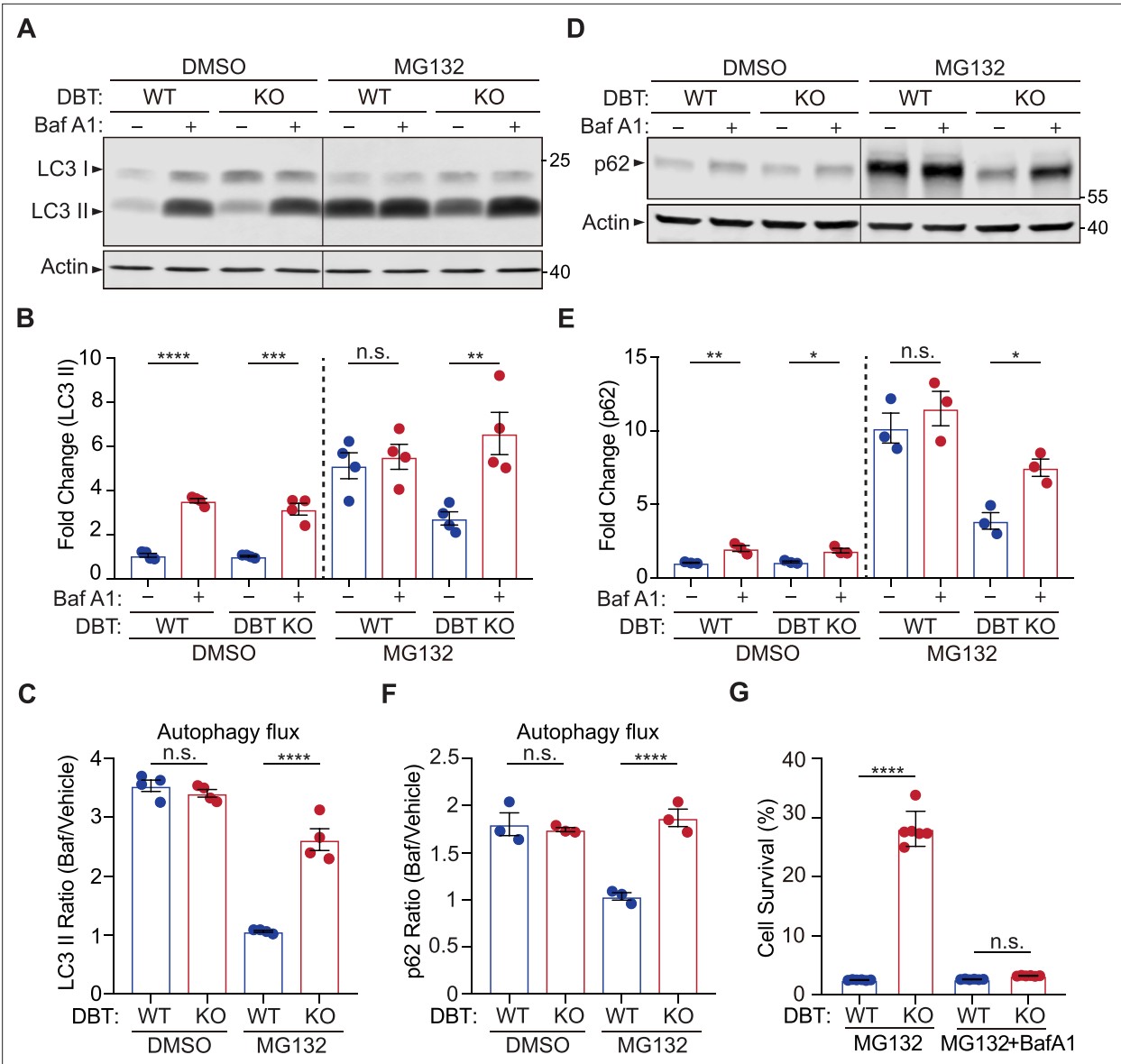

**Figure 3.** Loss of dihydrolipoamide branched chain transacylase E2 (DBT) preserves autophagic activities under proteasomal inhibition. (**A**) Immunoblot analysis of LC3II levels in wild-type (WT) and DBT knockout (KO) retinal pigment epithelium (RPE1) cells treated with MG132 (2 μM, 48 hr) and with or without Baf A1 (Bafilomycin A1, 100 nM, 4 hr). (**B**) Quantification of the LC3II levels in (**A**) (n=4). (**C**) Quantification of the autophagic flux as measured by the ratios of LC3II levels before and after the Baf A1 treatment (n=4). (**D**) Immunoblot analysis of p62 in WT and DBT KO cells treated with MG132 (2 μM, 48 hr) with or without Baf A1 (Bafilomycin A1, 100 nM, 4 hr). (**E**) Quantification of the p62 levels in (**D**) (n=3). (**F**) Quantification of the autophagic flux as measured by the ratios of p62 levels before and after the Baf A1 treatment (n=3). (**G**) Cell viability analysis with crystal violet staining was performed on WT and DBT KO RPE1 cells treated with MG132 and with or without Baf A1 (n=6). Error bars represent means ± SEM. 'n.s.', no significance; *p≤0.05; **p≤0.01; ***p≤0.001; ****p≤0.0001.

The online version of this article includes the following source data and figure supplement(s) for figure 3:

**Source data 1.** Original and uncropped blots for *Figure 3A*.

**Source data 2.** Original and uncropped blots for *Figure 3D*.

**Figure supplement 1.** Loss of dihydrolipoamide branched chain transacylase E2 (DBT) promotes cellular resistance to MG132-induced toxicity through the AMP-activated protein kinase (AMPK) signaling pathway.

**Figure supplement 1—source data 1.** Original and uncropped blots for *Figure 3—figure supplement 1A*.

**Figure supplement 1—source data 2.** Original and uncropped blots for *Figure 3—figure supplement 1B*.

**Figure supplement 1—source data 3.** Original and uncropped blots for *Figure 3—figure supplement 1D*.

**Figure supplement 1—source data 4.** Original and uncropped blots for *Figure 3—figure supplement 1E*.

significantly increased levels of BCAAs, in line with the notion that DBT is essential for BCAA catabolism (*Figure 4A*, left). Although MG132 treatment led to a decrease in the BCAA levels in the WT cells, the BCAA levels in the DBT KO cells remained elevated and showed no change upon the MG132 treatment (*Figure 4A*), confirming that the BCAA catabolism was completely blocked in the absence of DBT. Then we asked whether the accumulation of BCAAs played a role in the resistance of DBT KO cells to MG132 toxicity. When the BCAA levels were increased by 6–8 folds in the culture media of WT RPE1 cells, we did not observe any protection against MG132 toxicity (*Figure 4—figure supplement 1B*), indicating that the BCAA accumulation alone was not sufficient to protect the cells from toxicity associated with the proteasomal impairment.

The catabolism of BCAAs consists of two main steps (*Figure 4—figure supplement 1A*). The first step is mediated by BCAT, which is that enzyme converting BCAAs and α-ketoglutarate into branched-chain α-keto acids and glutamate. The second step is mediated by the BCKD enzyme complex, with DBT as one of its components. As DBT deficiency was found to protect cells from proteotoxicity, we further asked how the deficiency in other components of the BCAA catabolism could affect cell survival under proteotoxic stress. We generated knockout RPE1 cells lacking BCAT or BCKDHA, the latter being a component of the BCKD enzyme complex, disrupting the first and second steps of the BCAA catabolism, respectively. Depletion of BCAT or BCKDHA did not affect cell survival under normal conditions. However, the BCAT or BCKDHA knockout cells exhibited significant resistance to MG132-induced cytotoxicity (*Figure 4—figure supplement 1C*), similar to DBT knockout cells. Taken together, these findings demonstrate that it is not the accumulation of BCAAs but the blockade of the BCAA catabolism that leads to resistance to MG132-induced toxicity.

Since both the proteasome-mediated proteolysis and the BCAA catabolism can influence energy metabolism (*Ye et al., 2020*; *Szczepanowska and Trifunovic, 2021*), we examined the energetic status of WT and DBT KO RPE1 cells. Under normal conditions, the DBT KO cells showed ATP/ADP ratios comparable to those of WT cells. The MG132 treatment slightly decreased the ATP/ADP ratios in the WT cells; however, the MG132-treated DBT KO cells exhibited significantly decreased ATP/ADP ratios compared to the untreated DBT KO cells or the WT cells with or without the MG132 treatment (*Figure 4B*). Given that glucose is the primary source of ATP production in the cell, we sought to analyze the effects of increasing glucose concentrations on the sensitivity of DBT KO cells to MG132-induced toxicity. When WT cells were examined, increasing the glucose concentrations in the media from 17.5 mM to 70 mM or 140 mM did not affect the sensitivity of the WT cells to MG132 toxicity (*Figure 4—figure supplement 1D*). In contrast, when the cell survival of DBT KO cells was examined, the increased glucose levels significantly decreased the survival of DBT KO cells under MG132 treatment in a glucose-concentration-dependent manner (*Figure 4—figure supplement 1D*), in accordance with the notion that reduction in ATP levels contributed to the resistance of DBT KO cells to MG132 toxicity. Furthermore, glucose starvation rendered WT cells more resistant to MG132-induced toxicity (*Figure 5—figure supplement 1C*), consistent with the notion that reduction in ATP levels contributed to the cellular resistance to the toxicity. Together, these results indicate that loss of DBT combined with proteasomal inhibition led to a significant decrease in the ATP/ADP ratios, which correlated with the preservation of autophagic activities and the resistance to cell death induced by the proteasomal inhibition.

AMPK is a highly conserved sensor of cellular energy changes and is activated by increasing levels of AMP or ADP coupled with falling ATP (*Mihaylova and Shaw, 2011*). To assess the effect of DBT on AMPK signaling, we examined the activation of AMPK, as measured by the phosphorylation of its threonine 172 ($T^{172}$) (*Hardie, 2011*), in WT and DBT KO RPE1 cells. The levels of phosphorylated AMPK-$T^{172}$ were not significantly changed in the DBT KO cells under the resting condition; however, upon the MG132 treatment, the DBT KO cells exhibited significantly higher levels of phosphorylated AMPK-$T^{172}$ than the WT cells (*Figure 4C*). The total level of AMPK was not changed by loss of DBT or the MG132 treatment alone (*Figure 4C*). These data indicated that AMPK was activated in cells under the MG132-induced proteotoxic stress in the absence of DBT.

Next, we tested the role of AMPK in the resistance of DBT KO cells to MG132-induced toxicity. We knocked down AMPK by using specific shRNAs and examined the cell survival in the DBT KO cells with the MG132 treatment. The AMPK knockdown abolished the resistance of the DBT KO cells to the MG132-induced cytotoxicity. Under the MG132-treated conditions, the AMPK knockdown did not affect the survival of WT RPE1 cells; however, it significantly decreased the survival of DBT KO cells to

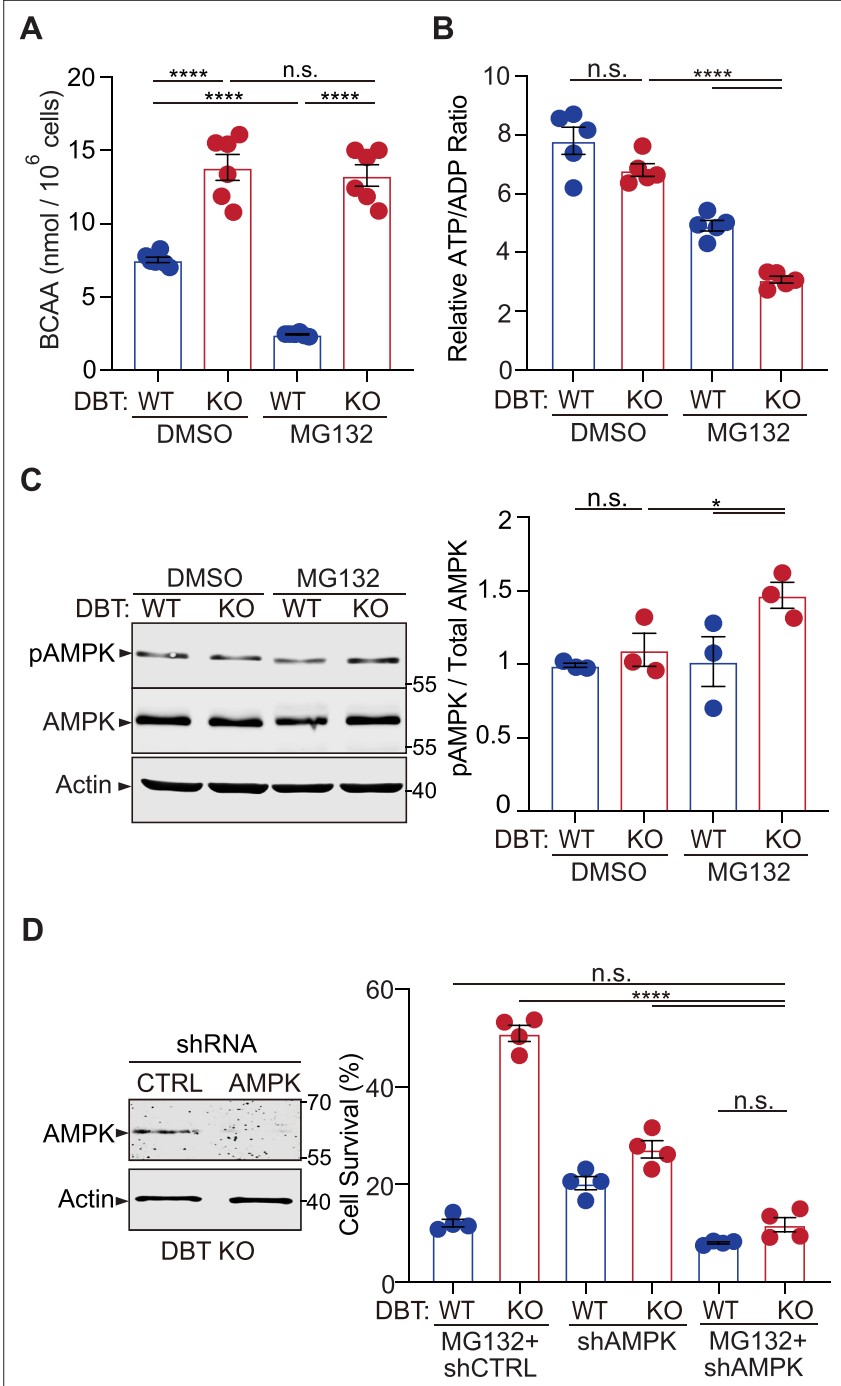

**Figure 4.** Loss of dihydrolipoamide branched chain transacylase E2 (DBT) activates AMP-activated protein kinase (AMPK) under proteasomal inhibition through energy regulation. (**A**) Intracellular branched-chain amino acid (BCAA) levels were measured in wild-type (WT) and DBT knockout (KO) retinal pigment epithelium (RPE1) cells treated with MG132 (2 μM, 48 hr) (n=6). (**B**) Intracellular ATP/ADP ratios were measured in WT and DBT KO RPE1 cells treated with MG132 (2 μM, 48 hr) (n=5). (**C**) Activation of AMPK in DBT KO RPE1 cells treated with MG132 (2 μM, 48 hr), as indicated by the increase in the levels of phosphorylated AMPK (n=3). (**D**) The knockdown of AMPK by specific small hairpin RNAs (shRNAs) abolished the protective effects of loss of DBT against MG132-induced toxicity in DBT KO RPE1 cells, as indicated by the cell viability measured with crystal violet staining (n=4). Error bars represent means ± SEM. 'n.s.', no significance; *p≤0.05; ****p≤0.0001.

The online version of this article includes the following source data and figure supplement(s) for figure 4:

**Source data 1.** Original and uncropped blots for *Figure 4C*.

*Figure 4 continued on next page*

*Figure 4 continued*

**Source data 2.** Original and uncropped blots for *Figure 4D*.

**Figure supplement 1.** The energy depletion but not branched-chain amino acid (BCAA) accumulation mediates the resistance of dihydrolipoamide branched chain transacylase E2 (DBT) knockout cells to MG132-induced proteotoxicity.

a level comparable to that of WT cells (*Figure 4D*). To confirm the role of the AMPK signaling in the cell survival phenotype of DBT KO cells, we applied an AMPK inhibitor, Compound C (*Zhou et al., 2001*), and found that it also abolished the protective effects of loss of DBT and rendered the DBT KO cells sensitive to the MG132 treatment (*Figure 5—figure supplement 1A and B*). Conversely, using an AMPK agonist EX229, also known as compound 991 (*Xiao et al., 2013*; *Lai et al., 2014*), we found that the AMPK agonist promoted the resistance to MG132-induced toxicity in both WT and DBT KO cells (*Figure 5—figure supplement 1D*). These results indicate that AMPK is a key player in the resistance of DBT KO cells against MG132-induced toxicity.

## Loss of DBT activates autophagy under proteotoxic stress via AMPK effectors

Since AMPK is a principal regulator of autophagy, we tested the role of AMPK in the autophagy activation in the DBT KO cells under MG132-induced proteotoxic stress. The knockdown of AMPK via shRNAs significantly reduced the LC3 protein levels, and quantification of LC3II levels before and after Bafilomycin A1-mediated inhibition of lysosomal degradation indicated that the autophagy flux was significantly reduced by the downregulation of AMPK (*Figure 5A*), consistent with the increased vulnerability of the DBT KO cells to the MG132-induced proteotoxic stress (*Figure 4D*).

The activation of AMPK promotes autophagy through several downstream effectors, including serine/threonine-protein kinase ULK1 and mTOR signaling (*Mihaylova and Shaw, 2011*). ULK1 is an important regulator of autophagosome biogenesis and can be activated by AMPK signaling (*Kim et al., 2011*). In the absence of MG132 treatment, there was no difference in the levels of total ULK1 protein or its activity, as measured by the phosphorylation at its serine 371 (S$^{371}$) site (*Figure 5B*). Upon the treatment with 2 µM of MG132 for 48 hr, there was little change in the ULK1 activity in WT RPE1 cells; however, the DBT KO cells exhibited a significant increase in the phosphorylation of ULK1-S$^{371}$ (*Figure 5B*), indicating that loss of DBT specifically induced the activation of ULK1 under the MG132 treatment, concomitant with the AMPK activation under this condition (*Figure 4C*). Conversely, the elevated expression of DBT significantly decreased the levels of phosphorylated ULK1 and phosphorylated AMPK in both WT and DBT KO cells (*Figure 3—figure supplement 1E*). These results indicate that loss of DBT specifically induces the activation of the AMPK-ULK1 signaling under MG132 treatment.

Activated AMPK also inhibits the mTOR signaling, which negatively regulates autophagy. AMPK is reported to regulate the mTOR signaling by phosphorylating TSC2 on serine 1387 (S$^{1387}$) (*Mihaylova and Shaw, 2011*). Phosphorylation of TSC2-S$^{1387}$ enhances the activity of TSC2 as a Rheb GAP, leading to the inhibition of mTOR. We found that the changes in the phosphorylation of TSC2-S$^{1387}$ were in accordance with that of AMPK-T$^{172}$ in the WT and DBT KO cells under the MG132-induced proteotoxic stress. There was little difference in the phosphorylation levels of TSC2 between the WT and DBT KO cells in the absence of MG132 treatment; however, under the treatment of 2 µM of MG132 for 48 hr, the DBT KO cells exhibited significantly higher levels of phosphorylated TSC2-S$^{1387}$, while the WT cells did not show any change in the phosphorylation levels of TSC2 (*Figure 5B*). These results suggest that the AMPK-TSC2-mTOR signaling is another mechanism through which loss of DBT leads to autophagic activation under proteasomal impairment.

Next, we modulated the activity of the mTOR signaling in the DBT KO cells to test its roles in the observed phenotypes of autophagic activation and resistance of MG132-induced cell toxicity. Since inhibition of mTOR signaling is associated with autophagic activation, as observed in the DBT KO cells, we targeted TSC1, a negative regulator of mTOR, via shRNAs to increase the mTOR signaling in the DBT KO cells. The increase in the mTOR activity as a result of TSC1 knockdown, as confirmed by the upregulated S6K phosphorylation (*Figure 5C*), led to a significant reduction in LC3II protein levels as well as in the autophagy flux measured by the LC3II ratios before and after Bafilomycin treatment, confirming that the autophagic activation in the DBT KO cells was dependent on the inhibition of

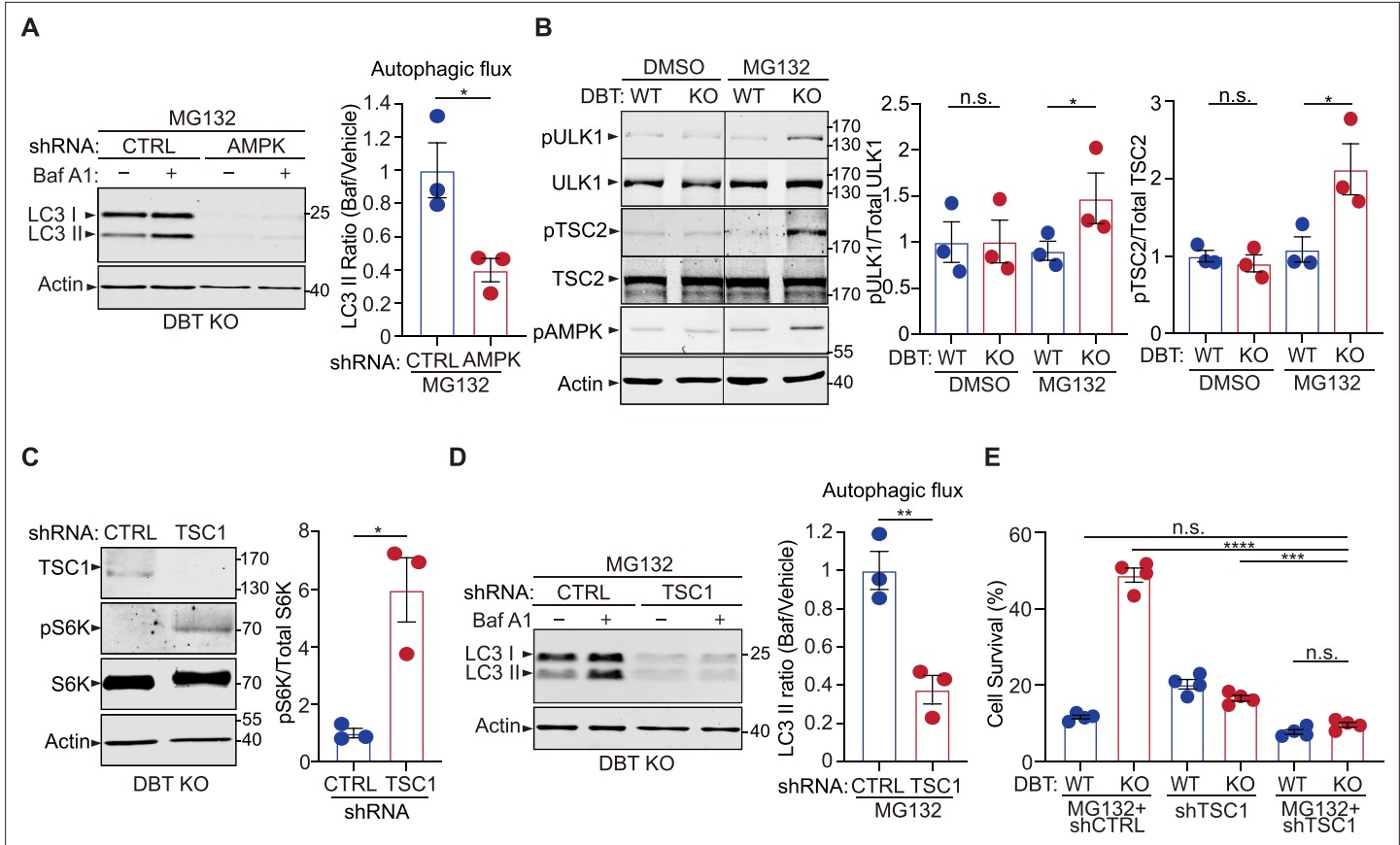

**Figure 5.** AMP-activated protein kinase (AMPK) downstream signaling enhances autophagy upon dihydrolipoamide branched chain transacylase E2 (DBT) deficiency and proteasomal inhibition. (**A**) Immunoblot analysis of the autophagy marker LC3II in DBT knockout (KO) retinal pigment epithelium (RPE1) cells after the knockdown of AMPK using small hairpin RNAs (shRNAs) versus non-targeting control shRNAs (CTRL), under MG132 treatment conditions (2 μM, 48 hr), with or without the Baf A1 treatment. The autophagic flux is measured by calculating the ratios of LC3II protein levels with the Baf A1 treatment to those without the Baf A1 treatment (n=3). (**B**) Immunoblot analysis of the AMPK downstream effectors that regulate mTOR activities, including ULK1 and TSC2, in WT and DBT KO RPE1 cells with or without treatment with MG132 (2 μM, 48 hr). The activities of these regulators are quantified by measuring the levels of phosphorylation of ULK1-S$^{371}$, TSC2-S$^{1387}$, and AMPK-T$^{172}$ (n=3). (**C**) Immunoblot analysis of the mTOR downstream marker S6K and its phosphorylation in DBT KO RPE1 cells after the knockdown of the negative regulator of mTOR, TSC1, using shRNAs versus control shRNAs (CTRL), under MG132 treatment conditions (2 μM, 48 hr) (n=3). (**D**) Immunoblot analysis of LC3II and quantification of the autophagic flux in DBT KO RPE1 cells after the knockdown of TSC1 under MG132 treatment conditions (2 μM, 48 hr) with or without the Baf A1 treatment (n=3). (**E**) Cell viability analysis with crystal violet staining of WT and DBT KO RPE1 cells after the knockdown of TSC1 using shRNAs versus control shRNAs (CTRL), under MG132 treatment conditions (2 μM, 48 hr) (n=4). Error bars represent means ± SEM. 'n.s.', no significance; *p≤0.05; **p≤0.01; ****p≤0.0001.

The online version of this article includes the following source data and figure supplement(s) for figure 5:

**Source data 1.** Original and uncropped blots for *Figure 5A*.

**Source data 2.** Original and uncropped blots for *Figure 5B*.

**Source data 3.** Original and uncropped blots for *Figure 5C*.

**Source data 4.** Original and uncropped blots for *Figure 5D*.

**Figure supplement 1.** Loss of dihydrolipoamide branched chain transacylase E2 (DBT) activates AMP-activated protein kinase (AMPK) and inhibits mTOR signaling.

**Figure supplement 1—source data 1.** Original and uncropped blots for *Figure 5—figure supplement 1A*.

**Figure supplement 1—source data 2.** Original and uncropped blots for *Figure 5—figure supplement 1E*.

mTOR signaling (*Figure 5D*). Furthermore, in accordance with the changes in autophagic activities, the knockdown of TSC1 abolished the enhanced survival phenotype of the DBT KO cells under the MG132 treatment (*Figure 5E*). To confirm the role of the mTOR signaling in the cell survival phenotype of DBT KO cells, we applied an mTOR agonist, MHY1485 (*Choi et al., 2012*), and found that it also abolished the protective effects of loss of DBT and rendered the DBT KO cells sensitive to the

MG132 treatment (*Figure 5—figure supplement 1E and F*). Conversely, treatment with an mTOR inhibitor, RAD001 or AZD8055 (*Beuvink et al., 2005*; *Willems et al., 2012*), increased the resistance of WT cells to MG132-induced toxicity (*Figure 5—figure supplement 1G*). These results demonstrate that the mTOR signaling plays a key role in the resistance of DBT KO cells to the proteasomal inhibition-induced toxicity.

## Loss of DBT suppresses the toxicity of mutant TDP-43 by promoting its clearance

Neurodegeneration-associated protein misfolding and aggregation often lead to impairment of the proteasome (*Ciechanover and Brundin, 2003*). ALS-linked mutant TDP-43 proteins, including the M337V variant, are prone to misfolding and aggregation, providing a disease-related molecular model for studying proteotoxicity (*Johnson et al., 2009*). We asked whether loss of DBT protects cells from mutant TDP-43-associated proteotoxicity. TDP-43$^{M337}$ or a GFP-like control protein Dendra2 was expressed in RPE1 cells for 48 hr, and the number of live cells was quantified by Calcein-in-AM staining. The WT cells expressing TDP-43$^{M337}$ showed significantly lower cell viability than those expressing the control protein, indicating that the WT cells were highly sensitive to TDP-43$^{M337}$-induced proteotoxicity (*Figure 6A*). However, the DBT KO cells exhibited significantly higher levels of cell survival than the WT cells when expressing TDP-43$^{M337}$, indicating strong protection by the loss of DBT. Furthermore, we asked whether loss of DBT would alleviate TDP-43$^{M337}$-induced proteotoxicity in mammalian neurons. We employed motor neurons differentiated from mouse embryonic stem cells (*Zhang et al., 2020*), which showed marked sensitivity to virally expressed TDP-43$^{M337}$ when compared to the control GFP protein, as shown by the significant neuronal loss with the TDP-43$^{M337}$ expression (*Figure 6B*). Importantly, the knockdown of DBT by specific shRNAs significantly rescued the neuronal loss with the TDP-43$^{M337}$ expression, as compared to the non-targeting control shRNAs (*Figure 6B*). These results demonstrate that loss of DBT resulted in robust protection against TDP-43-associated proteotoxicity.

Next, we asked whether the resistance of DBT KO cells to TDP-43$^{M337V}$-induced proteotoxicity was associated with any change in the turnover of the mutant protein. In immunoblotting analysis, TDP-43$^{M337V}$ transiently expressed in the DBT KO RPE1 cells showed substantially lower protein levels than that expressed in the WT cells (*Figure 6C*). To determine whether the decrease in the steady-state levels of TDP-43$^{M337V}$ protein was a consequence of increased protein degradation, we performed the cycloheximide chase assay to measure the turnover rate of the TDP-43$^{M337V}$ protein. As analyzed by immunoblotting of TDP-43$^{M337V}$ over a period of time after cycloheximide-induced inhibition of protein synthesis, the degradation of TDP-43$^{M337V}$ took place at a much faster rate in the DBT KO cells than in the WT cells (*Figure 6D*). The half-life of TDP-43$^{M337V}$ was estimated to be over 24 hr in the WT cells but only ~4.5 hr in the DBT KO cells. Furthermore, consistent with the faster turnover rate of TDP-43$^{M337V}$, the DBT KO cells exhibited a higher degree of autophagy flux than the WT cells, as measured by LC3II levels before and after the Bafilomycin treatment (*Figure 6E*). Additionally, the elevated expression of TDP-43$^{M337V}$ selectively increased the levels of phosphorylated AMPK in the DBT KO cells but not in the WT cells (*Figure 7—figure supplement 1*). These data indicate that loss of DBT promoted the clearance of TDP-43$^{M337V}$ and reduced its proteotoxicity through enhanced AMPK signaling and autophagic functions.

To study the TDP-43-associated proteotoxicity in an in vivo system, we employed a *Drosophila* model, which expresses the ALS-associated human mutant TDP-43$^{M337V}$ under the GMR-GAL4 driver and develops a rough-eye phenotype as a result of loss of photoreceptors and retinal degeneration in adult eyes (*Ritson et al., 2010*). By crossing with a transgenic strain with stable RNAi-induced reduction of the *Drosophila* homolog of human DBT, CG5599 (*St Pierre et al., 2014*), we found that the knockdown of the DBT homolog produced a significant rescue of the TDP-43$^{M337V}$-induced rough eye phenotype, resulting in a smoother eye appearance and increased pigmentation (*Figure 6F*). To confirm the observation, we generated a DBT KO *Drosophila* strain by mating Cas9 flies with DBT sgRNA flies and obtained a homozygous mutant with a 1 bp deletion in exon 2 of the *Drosophila* DBT gene, which yielded a null allele with a premature stop codon in exon 2 (*Figure 1—figure supplement 1E*). When we crossed the DBT KO flies with TDP-43$^{M337V}$ transgenic flies, a similar protection against the rough eye phenotype was observed (*Figure 6F*), indicating that loss of DBT protects against TDP-43-associated neurodegeneration.

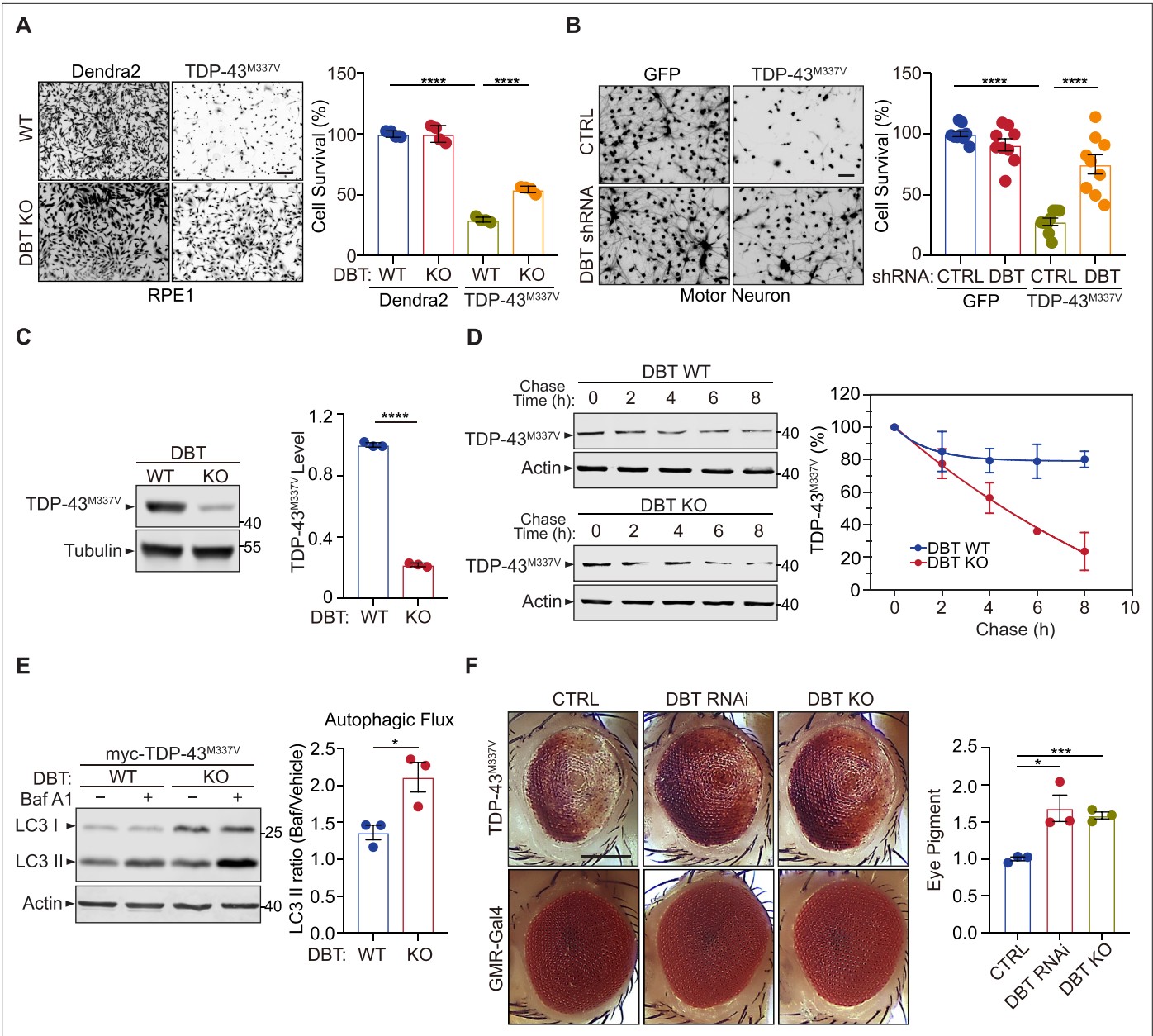

**Figure 6.** Loss of dihydrolipoamide branched chain transacylase E2 (DBT) protects against proteotoxicity of mutant TDP-43 in mammalian neurons and *Drosophila* models. (**A**) Cell toxicity of amyotrophic lateral sclerosis (ALS)-linked TDP-43^M337V expressed in wild-type (WT) and DBT knockout (KO) retinal pigment epithelium (RPE1) cells, with Dendra2 as a control, as measured by Calcein-AM staining (n=4). Scale bar, 300 μm. (**B**) Neuronal toxicity of TDP-43^M337V expressed in mouse embryonic stem (ES) cell-differentiated motor neurons with or without DBT small hairpin RNA (shRNA)-mediated knockdown, compared to that of GFP control, as measured by Calcein-AM staining (n=9). Scale bar, 300 μm. (**C**) TDP-43^M337V protein steady-state levels are significantly lower in DBT KO RPE1 cells than in WT control cells, as measured by immunoblot analysis (n=3). (**D**) The half-life of TDP-43^M337V protein as measured in cycloheximide chase assays is significantly shorter in the DBT KO cells than in WT RPE1 control cells (n=3 independent experiments; p=0.0326). (**E**) Immunoblot analysis of the autophagy marker LC3II and quantification of the autophagic flux in WT and DBT KO RPE1 cells transfected with myc-TDP-43^M337V with or without the Baf A1 treatment. The autophagic flux is measured by calculating the ratios of LC3II protein levels with the Baf A1 treatment to those without the Baf A1 treatment (n=3). (**F**) The reduction of DBT by RNAi or CRISPR led to strongly suppressed eye degeneration phenotypes in the TDP-43^M337V fly strain when compared with the control Luc RNAi (CTRL). The eye degeneration phenotypes were quantified by measuring the pigment content in adult eyes (n=3 independent groups with each containing fly heads from four males and fpur females). Scale bar, 100 μm. Error bars represent means ± SEM. *p≤0.05; ***p≤0.001; ****p≤0.0001.

The online version of this article includes the following source data and figure supplement(s) for figure 6:

**Source data 1.** Original and uncropped blots for *Figure 6C*.

*Figure 6 continued on next page*

*Figure 6 continued*

**Source data 2.** Original and uncropped blots for *Figure 6D*.

**Source data 3.** Original and uncropped blots for *Figure 6E*.

**Figure supplement 1.** Loss of dihydrolipoamide branched chain transacylase E2 (DBT) protects against proteotoxicity of polyQ in mammalian neurons and *Drosophila* models.

**Figure supplement 1—source data 1.** Original and uncropped blots for *Figure 6—figure supplement 1C*.

To extend the analysis of DBT-dependent proteotoxicity from TDP-43 to another disease-associated protein, we focused on the polyglutamine (polyQ) repeat, which is a misfolded protein domain associated with neurodegenerative diseases such as Huntington's disease (HD) and spinocerebellar ataxia (SCA) (*Ross and Poirier, 2004*). In cell survival assays, we found that WT RPE1 cells expressing polyQ showed significantly lower cell viability than DBT KO cells, indicating that loss of DBT protects against polyQ-induced proteotoxicity (*Figure 6—figure supplement 1A and B*). In immunoblotting analysis, we found that polyQ transiently expressed in the DBT KO cells exhibited substantially lower protein levels than those in WT cells (*Figure 6—figure supplement 1C and D*), suggesting that the resistance of the DBT KO cells to polyQ-induced proteotoxicity was associated with enhanced turnover of the polyQ protein. Additionally, we utilized a *Drosophila* model to confirm the protective effects of loss of DBT polyQ-induced toxicity in an in vivo system. Based on the analysis of the rough eye phenotype in the polyQ transgenic fly model, we observed significant protection against eye degeneration in both DBT RNAi and KO flies (*Figure 6—figure supplement 1E and F*). Together, these results demonstrate that the evolutionarily conserved regulatory effects of DBT on proteotoxicity are likely broad-spectrum.

## DBT is dysregulated in ALS patients

To understand whether the DBT-dependent regulation of proteotoxicity is relevant to human patients, we analyzed the spinal cord tissues from a cohort of ALS patients, most of whom have been observed to harbor TDP-43 proteinopathy in their postmortem exams of central nervous systems (*Table 1*). First, we carried out the immunoblotting analysis of spinal cord tissue extracts from 24 ALS patients and eight non-neurological controls. Although the immunoblot signals showed variability among the individuals, we found that DBT protein levels were significantly elevated in the majority of ALS cases (*Figure 7A* and *Figure 7—figure supplement 1*). Next, we performed immunofluorescence staining to analyze the quantity and distribution of DBT proteins in the human spinal cord tissue sections. In the control cases, DBT could be seen clearly expressed in motor neurons, which are characterized by their large soma and unique nuclear morphologies (*Figure 7B*). Compared with the controls, the DBT immunofluorescence signals showed much higher intensity and broader distribution across the spinal cord sections, suggesting elevated levels of DBT in neuronal soma and neurites (*Figure 7B*). Indeed, stronger signals of DBT immunofluorescence could be detected in large neuronal cell bodies in the ALS cases (*Figure 7B*). These observations suggest that the DBT-mediated regulation of protein quality control is compromised in the disease-relevant tissues of ALS patients.

## Discussion

The present study uncovers a major regulator of protein quality control through an unbiased genetic screen and demonstrates the signaling pathway through which DBT regulates autophagic activation under stress induced by proteasomal inhibition (*Figure 8*). The observations of DBT's robust regulatory effects on the proteotoxicity in models of neurodegenerative diseases and the upregulation of DBT in ALS patient tissues suggest that DBT is an important player in the pathogenesis of the disease. The findings could contribute to our better understanding of the regulation of protein quality control systems under stress or disease conditions.

DBT is a core component of the BCKD enzyme complex, which functions to break down BCAAs and provide acetyl-CoA or succinyl-CoA that can be used for energy production (*Brosnan and Brosnan, 2006*). Recent studies in *C. elegans* suggested that BCAA metabolism plays complicated roles in the regulation of the ubiquitin-proteasome system, with deficiency in components of the BCKD complex negatively impacting the ubiquitin-proteasome activity (*Ravanelli et al., 2022*). Surprisingly, we found

**Table 1.** The list of human patient's tissues.

Table: List of patient's tissues.

| Sample No. | Patient ID | Source | Clinical diagnosis | ALS pathology | Age of sampling | Gender | Region |
|---|---|---|---|---|---|---|---|
| 1 | 95 | TALS | CTRL | Non | 72 | M | SC-C |
| 2 | 103 | TALS | CTRL | Non | 22 | M | SC-C |
| 3 | 110 | TALS | CTRL | Non | 50 | M | SC-C |
| 4 | 108 | TALS | CTRL | Non | 72 | M | SC-C |
| 5 | 90015 | VABBB | CTRL | Non | 66 | M | SC-C |
| 6 | 90018 | VABBB | CTRL | Non | 82 | M | SC-C |
| 7 | 100012 | VABBB | CTRL | Non | 81 | F | SC-C |
| 8 | 120016 | VABBB | CTRL | Non | 63 | F | SC-C |
| 9 | AZ160030 | VABBB | ALS | Yes (TDP-43 pathology) | 65 | M | SC-C |
| 10 | AZ140006 | VABBB | ALS | Yes (TDP-43 pathology) | 74 | M | SC-C |
| 11 | 110011 | VABBB | ALS | Yes (TDP-43 pathology) | 83 | M | SC-C |
| 12 | 140008 | VABBB | ALS | Yes (TDP-43 pathology) | 75 | M | SC-C |
| 13 | AZ150001 | VABBB | ALS | Yes (TDP-43 pathology) | 65 | M | SC-C |
| 14 | AZ140021 | VABBB | fALS | Yes | 63 | M | SC-C |
| 15 | AZ150004 | VABBB | ALS | Yes (TDP-43 pathology) | 67 | M | SC-C |
| 16 | 130022 | VABBB | ALS | Yes (TDP-43 pathology) | 48 | M | SC-C |
| 17 | 130025 | VABBB | ALS | Yes (TDP-43 pathology) | 77 | M | SC-C |
| 18 | AZ140023 | VABBB | ALS | Yes (TDP-43 pathology) | 68 | M | SC-C |
| 19 | 130020 | VABBB | ALS | Yes (TDP-43 pathology) | 78 | M | SC-C |
| 20 | 130014 | VABBB | ALS | Yes (TDP-43 pathology) | 70 | M | SC-C |
| 21 | 100007 | VABBB | fALS | Yes (TDP-43 pathology) | 61 | M | SC-C |
| 22 | 100040 | VABBB | ALS | Yes | 88 | M | SC-C |
| 23 | 120015 | VABBB | ALS | Yes (TDP-43 pathology) | 58 | M | SC-C |
| 24 | 90003 | VABBB | fALS | Yes | 73 | M | SC-C |
| 25 | 90005 | VABBB | ALS | Yes (TDP-43 pathology) | 65 | M | SC-C |
| 26 | 90020 | VABBB | fALS | Yes | 49 | F | SC-C |
| 27 | 100002 | VABBB | ALS | Yes (TDP-43 pathology) | 63 | M | SC-C |
| 28 | AZ140017 | VABBB | ALS | Yes (TDP-43 pathology) | 66 | M | SC-C |
| 29 | 38 | TALS | sALS | Yes (C9orf72 HRE) | 34 | F | SC-C |
| 30 | 88 | TALS | sALS | Yes (C9orf72 HRE) | 59 | M | SC-C |
| 31 | 92 | TALS | fALS | Yes (C9orf72 HRE) | 72 | M | SC-C |
| 32 | MY9 | TALS | ALS | Yes (C9orf72 HRE) | 62 | F | SC-C |

Abbreviations: ALS (Amyotrophic Lateral Sclerosis), fALS (familial amyotrophic lateral sclerosis), sALS (sporadic amyotrophic lateral sclerosis), CTRL (Control), M (Male), F (Female), TALS (Target ALS Human Postmortem Tissue Core), VABBB (VA Biorepository Brain Bank), SC-C (Spinal Cord-Cervical), and HRE (hexanucleotide repeat expansion). The mean age of the controls is 63.5 years versus 69.4 years for the patients.

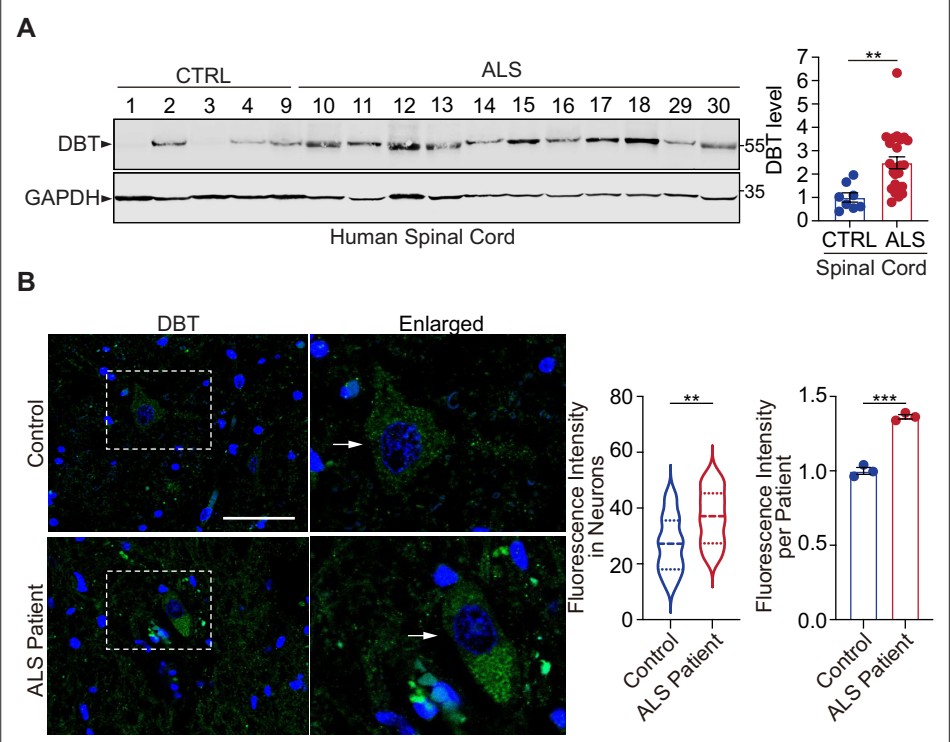

**Figure 7.** Dihydrolipoamide branched chain transacylase E2 (DBT) is abnormally upregulated in amyotrophic lateral sclerosis (ALS) patient neurons. (**A**) Immunoblot analysis of human spinal cord tissues from ALS patients and non-neurological controls (n=24 ALS cases and eight non-neurological control cases). (**B**) Fluorescent immunostaining against DBT in the spinal cords from ALS patients and an age-matched control cases indicates that the accumulation of DBT in the patient's neurons as identified by their morphological characteristics (n=23 neurons from three ALS cases and n=24 neurons from three control cases). Arrows point to representative motor neurons. Scale bar, 50 µm. Error bars represent means ± SEM. **p≤0.01.

The online version of this article includes the following source data and figure supplement(s) for figure 7:

**Source data 1.** Original and uncropped blots for *Figure 7A*.

**Figure supplement 1.** Analyses of the resistance of dihydrolipoamide branched chain transacylase E2 (DBT) knockout (KO) cells to TDP-43 toxicity and the increased DBT protein levels in amyotrophic lateral sclerosis (ALS) patients' tissues.

**Figure supplement 1—source data 1.** Original and uncropped blots for *Figure 7—figure supplement 1A*.

**Figure supplement 1—source data 2.** Original and uncropped blots for *Figure 7—figure supplement 1B*.

in the present study that mammalian cells deficient in DBT exhibited a remarkable degree of resistance to cell death from proteotoxicity induced by the proteasomal inhibition. In accordance with the enhanced cellular survival under the proteotoxic stress induced by proteasomal inhibition in cells lacking DBT, the DBT-deficient cells exhibited a remarkable capacity to prevent the accumulation of poly-ubiquitinated proteins as a result of the proteasomal inhibition. Although proteasomal inhibition itself may activate autophagy as an adaptive cellular response in wild-type cells during the early stages of stress, we have observed diminished autophagic activities after the persistent stress, likely due to the overloaded burden of proteotoxicity from proteasomal inhibition. The efficient clearance of the poly-ubiquitinated proteins despite the proteasomal inhibition in the DBT-deficient cells was attributable to a robust preservation of autophagic activities in the absence of DBT.

Conventionally, elevated levels of BCAAs are associated with inhibition of autophagy due to their ability to activate mTORC1 (*Zhenyukh et al., 2017*), the nutrient sensor that negatively regulates autophagy (*Dunlop and Tee, 2014*). While the expected increase in BCAAs was observed following the loss of DBT, our findings indicate that additional factors beyond BCAAs play a role in the regulation of autophagy, since DBT-deficient cells exhibited improved preservation of autophagic activities under the conditions of proteasomal inhibition compared to WT cells. Consistent with the preservation of

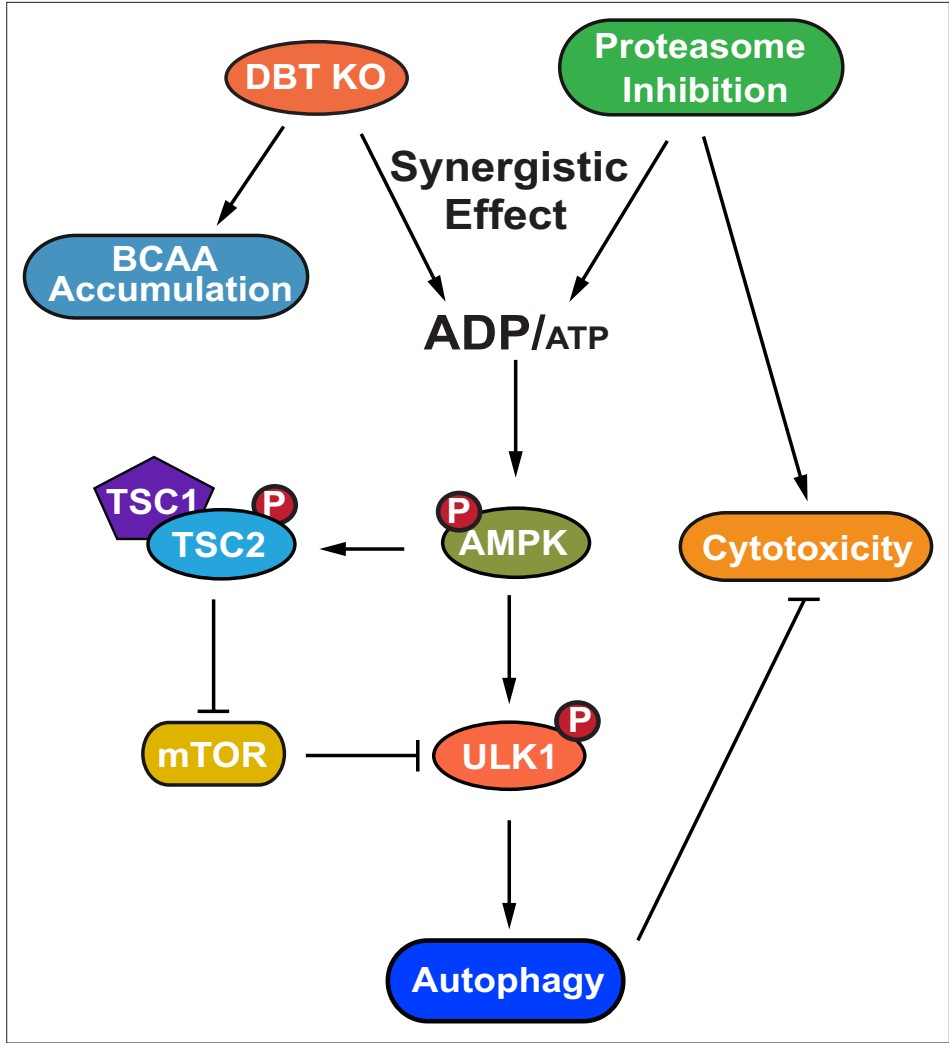

**Figure 8.** A model for the dihydrolipoamide branched chain transacylase E2 (DBT)-AMPK-autophagy signaling pathway. A working model of the mechanism through which DBT acts as a metabolic switch for the maintenance of protein homeostasis through activation of autophagy under the condition of proteasomal inhibition. The loss of DBT leads to the accumulation of BCAAs as a result of the blocked catabolism of these amino acids, which tilts the balance of intracellular energy and reduces the levels of ATP, under the condition of proteasomal inhibition. This energy imbalance triggers the activation of AMPK, which then promotes autophagy through its regulation of mTOR and ULK1. BCAA, branched-chain amino acid; ATP, adenosine triphosphate; ADP, adenosine diphosphate; AMPK, AMP-activated protein kinase; TSC1, TSC complex subunit 1; TSC2, TSC complex subunit 2; ULK1, unc-51 like autophagy activating kinase 1.

autophagic activities, the DBT-deficient cells showed significant activation of AMPK signaling under the conditions of proteasomal inhibition. The activation of AMPK, as a synergistic effect of proteasomal inhibition and DBT deficiency, is driven by the depletion of intracellular ATP under the combined conditions. The ATP-producing OXPHOS complexes are sensitive to the regulation of protein quality control, as its many components are regulated by proteasomal degradation (*Szczepanowska and Trifunovic, 2021*), consistent with our observation of lower ATP levels upon treatment with the proteasome inhibitor. We also found that the DBT deficiency decreased the intracellular ATP levels under the condition of proteasomal inhibition, suggesting that DBT-dependent BCAA catabolism is a critical source of energy production under the condition of proteasomal inhibition. Furthermore, the net outcome of better preservation of autophagic activities, upon inhibition of both proteasomal degradation and BCAA catabolism, underlies the important roles of the energy metabolism and the cognate AMPK signaling, which regulates autophagy regulators including mTOR and ULK1 (*Alers*

*et al., 2012*), as the driving force for the autophagic activation under these conditions. Although cells have complicated networks that regulate metabolism and proteostasis, the present study has revealed the pivotal role of DBT as a master switch for protein quality control and cell survival under proteotoxic stress such as that induced by the proteasomal inhibition.

The newly recognized function of DBT in regulating proteotoxicity is further demonstrated through the protective effects of DBT deficiency in cellular and *Drosophila* models of TDP-43 proteotoxicity. Together with the observations that DBT exhibits aberrant accumulation in ALS patients' spinal cord tissues, these results suggest that DBT may modulate proteotoxicity in neurodegenerative diseases. Both DBT and BCAAs play important roles in metabolism important for physiology and diseases. Defective BCAA catabolism is associated with metabolic disorders, such as maple syrup urine disease, insulin resistance, and diabetes (*Yoon, 2016*; *Holeček, 2018*). Abnormal regulation of the levels and metabolism of BCAAs has been implicated in a variety of other human pathological conditions, including Alzheimer's disease, cancer, and liver diseases (*Dimou et al., 2022*). The role of DBT as a critical regulator of protein quality control through metabolic and energetic controls under proteotoxic stress, as revealed in the present study, opens a new avenue for understanding the complex regulation of protein homeostasis in human health and pathologies.

## Methods

### DNA plasmids

The human DBT cDNA plasmid was obtained from the Ultimate ORF collection (Thermo Fisher). Synonymous mutations in the DBT cDNA were generated with Q5-site direct mutagenesis (NEB, E0554S) by replacing CACTTCCTGAAAACAACTGC with CATTTTTTAAAGACGACCGC in exon 1. The resulting variant DBT' was subcloned into the pDEST plasmid using the Gateway system (Thermo Fisher). For mammalian expression, human TDP-43$^{M337V}$ has subcloned into the pLenti CMV Puro DEST (W118-1, Addgene) plasmid as previously described (*Zhang et al., 2014*).

### Viruses

For lentivirus production, HEK293T cells were cultured under standard conditions and co-transfected overnight with a lentiviral transfer plasmid containing the insert of interest, the psPAX2 packaging plasmid (Addgene #12260), and the pMD2.G envelope plasmid (Addgene #12259) using Lipofect-amine 2000 (ThermoFisher #11668019). The lentiviral particle-containing medium was collected 72 hr after transfection, filtered through a 0.45 µm PVDF membrane (Millipore Sigma HVHP02500), mixed with PEG8000 at a 3:1 ratio, incubated with constant rocking at 60 rpm for 4 hr at 4°C, and centrifuged at 1600 × g at 4 °C for 1 hr. The supernatant was removed, and the viral pellet was resuspended in PBS, aliquoted, and stored at −80 °C. The TDP-43$^{M337V}$ and GFP HSVs were obtained from Gene Delivery Technology Core, Massachusetts General Hospital.

### Mammalian cell lines, transfections, and drug treatments

Human retinal pigment epithelial (RPE1, hTERT-RPE1, ATCC CRL-4000) cells were grown in Dulbecco's modified Eagle's medium/Nutrient Mixture F-12 (DMEM/F12, Life Technologies, 10565–018) supplemented with 10% fetal bovine serum (FBS) and 10 µg/mL hygromycin B (Corning, 30–240-CR) at 37 °C with 5% CO2. DBT knockout in RPE1 was achieved by infecting cells with viruses derived from pLenti-CRISPR v2 harboring the DBT-specific gRNA (5'- CACTTCCTGAAAACAACTGC-3'), and a population of puromycin-selected cells were used. Mouse embryonic fibroblasts (MEFs, ATCC, SCRC-1008) and human embryonic kidney 293 (HEK293) cells (ATCC, CRL-3216) were grown in DMEM supplemented with 10% FBS. The cell lines used in this study tested negative for mycoplasma contamination using the Mycolor One-step Mycoplasma Detector Kit (Vazyme) and were authenticated by short tandem repeat analysis. Transfection of mammalian cells was performed using Lipofectamine 2000 (Invitrogen). Briefly, 2 µg of the DNA plasmids and 4 µl of Lipofectamine 2000 were mixed in 500 µl Opti-MEM I (Invitrogen) and applied to RPE1 cell in 2 ml DMEM supplemented with 10% FBS. After two days, cells were lysed for analysis. MG132 was dissolved at 20 mM in dimethylsulfoxide (DMSO) and used at 2 µM for 48 hr, bortezomib dissolved at 2 mM in DMSO and used at 0.25 µM for 96 hr, Bafilomycin A1 dissolved at 400 µM in DMSO and used at 200 nM for 4 hr, MHY1485 dissolved at 5 mM in DMSO and used at 2 µM for 48 hr, RAD001 dissolved at 10 mM in DMSO and used at 50 nM

for 48 hr, AZD8055 dissolved at 10 mM in DMSO and used at 20 nM for 48 hr, EX229 dissolved at 10 mM in DMSO and used at 10 μM for 48 hr, and Vinblastine was dissolved at 10 mM in DMSO and used at 10 μM for 4 hr. All drugs were diluted in DMEM/F12 before the cell treatments. For glucose starvation, cells were cultured with fresh DMEM/F12 for 2 hr and then with glucose-free DMEM/F12 containing 10% dialyzed FBS for 48 hr. DBT knockdown in MEFs was achieved by infecting cells with viruses derived from pLKO-1 harboring the DBT shRNAs (TRCN0000099376, TRCN0000099377, and TRCN0000099378, Dharmacon) and a population of puromycin-selected cells were used. TSC1 or AMPK knockdown in RPE1 cells were achieved by infecting cells with viruses derived from pLKO-1 harboring the shRNAs against TSC1 (TRCN0000010453 and TRCN0000039734, Sigma) or AMPK (TRCN00000196482 and TRCN00000219690, Sigma), respectively, and a population of puromycin-selected cells were used.

## CRISPR-Cas9 gene editing and genome-wide screening

The specific gRNA sequences were selected by using the CRISPR design tool from Benchling, Inc The gRNAs were cloned into the gRNA/Cas9-expressing vector pLenti-CRISPR v2, conferring resistance to puromycin (Addgene 52961). After cell transduction with the lentiviruses expressing the Cas9/gRNAs, single-cell colonies were isolated based on puromycin resistance. The resulting cell lines were verified for their genotypes by sequencing the targeted locus or probing the targeted protein through immunoblot analysis.

The genomic CRISPR/Cas9 library with six sgRNA per gene was packaged into lentivirus and infected into RPE1 cells. Briefly, $6 \times 10^7$ *RPE1* cells were infected with the human sgRNA library (Human GeCKO v2 Library Cat#1000000048, Addgene) at an MOI of ~0.3. The infected cells were selected with 10 μg/ml puromycin for 7 days. After the puromycin selection, the infected RPE1cells were treated with MG132 at a concentration of 2 μM for 4 days, followed by recovery for another 4 days. After three rounds of treatments and recovery, the surviving resistant colonies were harvested, with the genomic DNA isolated followed by PCR amplification and Sanger sequencing of the sgRNA cassette.

## Cell viability assay using crystal violet or Calcein-AM

Cell viability was measured using crystal violet staining. Briefly, $5 \times 10^4$ RPE1 cells were seeded into a 24-well plate and treated with different concentrations of MG132 or bortezomib, or with the same volume of DMSO as solvent control. Cells were washed with 1 x PBS, fixed with 4% paraformaldehyde in PBS for 15 min, and stained with 0.1% crystal violet for 20 min at room temperature. For quantification, the crystal violet dye was extracted with 0.5 ml 10% acetic acid at room temperature for 20 min and the optical densities were measured at 570 nm using a Synergy H1 Hybrid Reader (BioTek).

Live-cell staining probe Calcein-AM (Invitrogen) was used for determining cell viability where the fluorescence intensity is proportional to the number of viable cells. Following various experimental treatments, $5 \times 10^4$ RPE1 cells were seeded into a 24-well plate and treated with MG132 (2 μM, 72 hr) or bortezomib (0.25 μM, 72 hr), or with the same volume of DMSO as a solvent control. Cells were washed with PBS and incubated with 1 μM Calcein-AM for 30 min in the dark at 37 °C under 5% $CO_2$. The fluorescent image was viewed with a Nikon TS100 fluorescence microscope, and a Synergy H1 Hybrid Reader (BioTek) with a custom filter (485 nm and 535 nm) was used to record the relative florescence intensity (RFI). RFI from the plate reader was used to calculate the ratios or fold changes of drug-treated groups over the control condition.

## Proteasome activity assay

The proteasome activity assay was performed using a Proteasome-Glo Chymotrypsin-Like Cell-Based Assay kit (Promega). Briefly, $1 \times 10^4$ cells were seeded into a 96-well culture plate. After 24 hr of MG132 or bortezomib treatment, a pre-mixed assay buffer containing substrates and luciferin detection reagents was added in equal volume to the sample and incubated at room temperature for 10 min. The supernatant was transferred to optiplate-96 (white; PerkinElmer) and luminescence was recorded using a fluorescent plate reader (Synergy H1, BioTek).

## Immunoblotting

Cells were washed twice with 1 X PBS and then lysed and harvested on ice in RIPA buffer (50 mM Tris-HCl, pH 7.6; 150 mM NaCl; 1% NP-40; 1% SDS; 100 mM sodium fluoride; 17.5 mM β-glycerophosphate;

0.5% sodium deoxycholate; and 10% glycerol). The RIPA buffer was supplemented with an EDTA-free protease inhibitor cocktail (Roche): phosphatase inhibitor cocktail (Roche), 1 µM phenylmethanesulfonyl fluoride, and 2 µM sodium orthovanadate. Cell lysates were kept cold on ice, pulse-sonicated for 10 min, and then centrifuged at 12,000 g at 4 °C for 10 min. The protein concentrations were determined using the bicinchonic acid (BCA) assay. Equal amounts of proteins from total cell lysates were resolved by SDS-PAGE and transferred to nitrocellulose membranes (Millipore). The blots were blocked with 5% w/v BSA and 0.05% $NaN_3$ in TBST and incubated with primary antibodies at 4 °C overnight, then finally incubated with appropriate secondary antibodies. The primary antibodies included anti-DBT (ab151991, Abcam); anti-β-actin (sc-47778, Santa Cruz); anti-PARP (9542, Cell Signaling); anti-Cleaved Caspase3 (9664, Cell Signaling); anti-ubiquitin (58395, Cell Signaling); anti-LC3B (3868, Cell Signaling and L7543, Sigma); anti-SQSTM1/p62 (5114, Cell Signaling); anti-GAPDH (5174, Cell Signaling); anti-AMPK (5832, Cell Signaling); anti-p-AMPK (Thr172) (2535, Cell Signaling); anti-TSC2 (2880, Cell Signaling); anti-TSC2 (Ser1387) (5584, Cell Signaling); anti-ULK1 (8054, Cell Signaling); anti-ULK1 (Ser317) (37762, Cell Signaling); anti-mTOR (2983, Cell Signaling); anti-mTOR (Ser2448) (5536, Cell Signaling); anti-P70S6K (5707, Cell Signaling); anti-P70S6K (Thr389) (9205, Cell Signaling); anti-TDP-43 (10782–2-AP, ProteinTech); and anti-Flag (F3165, Sigma). Proteins were visualized using Li-COR anti-mouse and anti-rabbit 680 and 800 fluorescent antibodies, and the images were captured with an Odyssey imager and analyzed with the Image Studio software (Licor).

## Immunofluorescence staining and detection of protein aggregates

For immunofluorescence staining, $8 \times 10^4$ cells were grown on polyethylenimine (PEI)-coated coverslips. The cells were fixed with 4% paraformaldehyde for 15 min at room temperature, and incubated in a blocking solution (5% normal goat serum, 0.1% Triton-X 100 in 1 X PBS) for 1 hr at room temperature. The coverslips were incubated with primary antibodies (anti-Ubiquitin ab7780, Abcam; anti-Ubiquitin BML-PW8810, Enzo; or anti-LC3B, 14600–1-AP, Proteintech) at 4 °C overnight, and then washed three times with PBS before being incubated with fluorochrome-conjugated secondary antibodies (anti-rabbit, Alexa Fluor 594; Invitrogen, Carlsbad, CA, 1:400) for 2 hr at room temperature. After 3–5 times of washes with PBS, the coverslips were mounted onto microscope slides using a mounting medium containing DAPI (P36931, Thermo Fisher).

Protein aggregates were detected using the ProteoStat Aggresome detection kit (ENZ-51035, Enzo) with modifications. Briefly, $8 \times 10^4$ cells were grown on polyethylenimine (PEI)-coated coverslips and treated with MG132 (2 µM, 72 hr) or the same volume of DMSO as a solvent control. Cells were washed twice with 1 X PBS, fixed with 4% paraformaldehyde for 30 min, and incubated with permeabilizing solution (0.5% Triton X-100, 3 mM EDTA, pH 8.0) for 30 min at room temperature. After being fixed and permeabilized, cells were then labeled with the ProteoStat reagent for 30 min at room temperature. After 3–5 times of washes with PBS, the coverslips were mounted onto microscope slides using a mounting medium containing DAPI (P36931, Thermo Fisher).

## BCAA measurement

Intracellular levels of BCAAs were analyzed using the BCAA Assay Kit (ab83374, Abcam) based on a colorimetric assay measuring the concentrations of molecules specifically derived from BCAAs in the assay. Briefly, $1 \times 10^6$ cells were seeded into a six-well plate and treated with MG132 or the solvent control DMSO. Following overnight incubation, the cells were washed carefully with PBS, and then lysed with 100 µl BCAA assay buffer. Following centrifugation at 15,000 x g for 10 min to remove cell debris and other insoluble materials, the samples were transferred into a 96-well assay plate and mixed with the reaction buffer, and then the OD at 450 nm was measured in a microplate reader (BioTek, Synergy H1).

## ATP/ADP measurement

The ATP/ADP ratio was analyzed using an ADP/ATP Ratio Assay Kit (MAK135, Sigma), which converts ATP and ADP into their respective intermediates and measures their concentrations in a bioluminescent assay. Cells were directly cultured in the assay microplate. Following overnight incubation, the culture medium in the assay microplate was replaced with the ATP reagent, and after an additional 1 min incubation, the ATP signal was measured on a plate reader (BioTek, Synergy H1). In the second

step, ADP was converted into ATP through an enzyme reaction and its signal was then measured. The ATP/ADP ratio was calculated according to the manufacturer's instructions.

## Mouse motor neuron differentiation

Mouse embryonic stem (mES) cells, provided by Shelly E. Sakiyama-Elbert laboratory at Washington University in St. Louis and authenticated by short tandem repeat analysis, were cultured and differentiated into motor neurons as previously described (*McCreedy et al., 2014*). Briefly, $2\times10^6$ mES cells were cultured in suspension in 10 ml DFK5 medium for 2 days, followed by incubation in DFK5 medium containing 2 µM retinoic acid and 0.6 µM smoothened agonist for 4 days. Differentiating embryonic bodies were dissociated, plated into a laminin-coated 12-well plate ($5\times10^5$), and incubated in DFK5 medium with GDNF, BDNF, and NT-3 (5 ng/ml each) for 24 hr. After 24 hr, the medium was replaced with DFKNB medium supplemented with 1 X B27, GDNF, BDNF, and NT-3.

Before changing the medium, cells were infected with a lentivirus construct expressing DBT shRNAs for 6 hr in 4 µg/ml polybrene. After a 6 hr infection, the medium was replaced with DFKNB supplemented with 1 X B27, GDNF, BDNF, and NT-3 for 48 hr. After 48 hr, cells were infected with herpes simplex viruses expressing TDP-43$^{M337V}$ or GFP in 4 µg/ml polybrene for 6 hr, as previously described (*Gupta et al., 2017*). After infection, motor neurons were incubated in a DFKNB medium supplemented with 1 X B27, GDNF, BDNF, and NT-3 for an additional 48 hr. On the last day, motor neurons were subjected to a cell survival assay using the Calcein AM method as described above.

## *Drosophila* strains and assays

Flies were reared and crossed on standard yeast agar cornmeal medium at 25 °C. The *Drosophila* transgenic strain carrying GAL4-inducible human TDP-43$^{M337V}$ (hTDP-43$^{M337V}$) (*Ritson et al., 2010*) was recombined with GMR-GAL4. The GMR-GAL4, hTDP-43$^{M337V}$/TM3 fly strains were then crossed to RNAi transgenic strains. The following *Drosophila* strains were obtained from the Bloomington Stock Center or Vienna Stock Center: RNAi strains including Luc (y1 v1; P{TRiP. JF01355}attP2); DBT (P{TRiP.HMS00663}attP2) and DBT (P{KK102355}VIE-260B); CRISPR knockout strains including Cas9 (y[1] sc[*] v[1] sev[21]; P{y[+t7.7] v[+t1.8]=nos-Cas9.R}attP40) and DBT sgRNA (y[1] sc[*] v[1] sev[21]; P{y[+t7.7] v[+t1.8]=TKO.GS01842}attP40). They were transferred to freshly made food every 2–3 d. Flies were aged for 21 d, and their eye morphology was examined. Changes in pigmentation, ommatidial structure, and glossiness phenotypes were monitored for enhancement or suppression. The pigment content was detected at 488 nm and the background corrected at 600 nm using fly head protein extracts as previously described (*Periz et al., 2015*; *Lu et al., 2019*). For the generation of CRISPR knockout flies, DBT sgRNA flies were crossed with Cas9 flies, then the filial generation flies were analyzed with Sanger sequencing of the genomic locus targeted by the sgRNA. A homozygous mutation was confirmed, where a 1 bp deletion in exon 2 resulted in a premature stop codon in exon 2 (*Figure 1—figure supplement 1E*).

## Statistical analysis

The statistical analyses were performed with Student's t-tests for two-group comparisons and one-way ANOVA with the Tukey post hoc test for multiple group comparisons using the Graphpad Prism software. The sample size 'n' represents biological replicates unless otherwise indicated. p-values less than 0.05 were considered statistically significant.

## Acknowledgements

This work was supported by grants from NIH (NS089616, NS110098, NS074324, and NS128494), Walder Foundation Endowment, Packard Center for ALS Research at Johns Hopkins, Target ALS Foundation, and ALS Association. We would like to thank Robert Kalb and Rachael Neve for viral expression vectors, Ian Robey at the VA Biorepository Brain Bank (VA merit review BX002466), Lyle Ostrow, Kathleen Wilsbach, and Kathryn Gallo at the Johns Hopkins ALS Postmortem Tissue Core, and the Target ALS Multicenter Postmortem Tissue Core for providing patient tissues, and the members of Wang lab for discussion.

## Additional information

### Funding

| Funder | Grant reference number | Author |
| --- | --- | --- |
| National Institutes of Health | NS089616 | Jiou Wang |
| National Institutes of Health | NS110098 | Jiou Wang |
| National Institutes of Health | NS074324 | Jiou Wang |
| National Institutes of Health | NS128494 | Jiou Wang |
| Packard Center for ALS Research at Johns Hopkins | | Jiou Wang |
| Target ALS | | Jiou Wang |
| The ALS Association | | Jiou Wang |

The funders had no role in study design, data collection, and interpretation, or the decision to submit the work for publication.

### Author contributions

Ran-Der Hwang, Conceptualization, Validation, Investigation, Writing – original draft, Designed the studies, Wrote the paper with inputs from G Periz and TZ, Designed most of the experiments; YuNing Lu, Conceptualization, Validation, Investigation, Writing – original draft, Writing – review and editing, Designed the studies, Wrote the paper with input from G Periz and TZ, Designed most of the experiments, performed the mESC culturing and neuronal differentiation, and performed the *Drosophila* experiments; Qing Tang, Validation, Investigation, Performed the mESC culturing and neuronal differentiation; Goran Periz, Investigation, Writing – original draft, Performed the *Drosophila* experiments, and wrote the paper; Giho Park, Validation, Investigation, Performed the BCAA analysis; Xiangning Li, Validation, Investigation, Writing – review and editing, Performed the proteasome immunoblot analysis; Qiwang Xiang, Investigation, Visualization, Performed the IF staining; Yang Liu, Investigation, Visualization, Performed the IHC staining; Tao Zhang, Investigation, Writing – review and editing, Wrote the paper; Jiou Wang, Conceptualization, Resources, Validation, Investigation, Writing – original draft, Project administration, Writing – review and editing, Designed the studies, Deigned all experiments, Supervised the project

### Author ORCIDs

YuNing Lu ⬚ https://orcid.org/0000-0002-8176-2108
Jiou Wang ⬚ https://orcid.org/0000-0001-9115-8708

Reviewer #1 (Public review): https://doi.org/10.7554/eLife.91002.4.sa1
Reviewer #2 (Public review): https://doi.org/10.7554/eLife.91002.4.sa2
Author response https://doi.org/10.7554/eLife.91002.4.sa3

## Additional files

### Supplementary files

• MDAR checklist

### Data availability

We have provided all the original and uncropped blots in the source data files.

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
