## [Editor Report · eLife assessment]

This **important** study discovered DBT as a novel gene implicated in the resistance to MG132-mediated cytotoxicity and potentially also in the pathogenesis of ALS and FTD, two fatal neurodegenerative diseases. The authors provided **convincing** evidence to support a mechanism by which loss of DBT suppresses MG132-mediated toxicity via promoting autophagy. This work will be of interest to cell biologists and biochemists, especially in the FTD/ALS field.

---

## [Referee Report · Reviewer #1 (Public review)]

Summary:

Through an unbiased genomewide KO screen, the authors identified loss of DBT to suppress MG132-mediated death of cultured RPE cells. Further analyses suggested that DBT reduces ubiquitinated proteins by promoting autophagy. Mechanistic studies indicated that DBT loss promotes autophagy via AMPK and its downstream ULK and mTOR signaling. Furthermore, loss of DBT suppresses polyglutamine- or TDP-43-mediated cytotoxicity and/or neurodegeneration in fly models. Finally, the authors showed that DBT proteins are increased in ALS patient tissues, compared to non-neurological controls.

Strengths:

The idea is novel, the evidence is convincing, and the data are clean. The findings have implications for human diseases.

Weaknesses:

None.

---

## [Referee Report · Reviewer #2 (Public review)]

Summary:

Hwang, Ran-Der et al utilized a CRISPR-Cas9 knockout in human retinal pigment epithelium (RPE1) cells to evaluate for suppressors of toxicity by the proteasome inhibitor MG132 and identified that knockout of dihydrolipoamide branched chain transacylase E2 (DBT) suppressed cell death. They show that DBT knockout in RPE1 cells does not alter proteasome or autophagy function at baseline. However, with MG132 treatment, they show a reduction in ubiquitinated proteins but with no change in proteasome function. Instead, they show that DBT knockout cells treated with MG132 have improved autophagy flux compared to wildtype cells treated with MG132. They show that MG132 treatment decreases ATP/ADP ratios to a greater extent in DBT knockout cells, and in accordance causes activation of AMPK. They then show downstream altered autophagy signaling in DBT knockout cells treated with MG132 compared to wild-type cells treated with MG132. Then they express the ALS mutant TDP43 M337 or expanded polyglutamine repeats to model Huntington's disease and show that knockdown of DBT improves cell survival in RPE1 cells with improved autophagic flux. They also utilize a *Drosophila* models and show that utilizing either a RNAi or CRISPR-Cas9 knockout of DBT improves eye pigment in TDP43M337V and polyglutamine repeat-expressing transgenic flies. Finally, they show evidence for increased DBT in postmortem spinal cord tissue from patients with ALS via both immunoblotting and immunofluorescence.

Strengths:

This is a mechanistic and well-designed paper that identifies DBT as a novel regulator of proteotoxicity via activating autophagy in the setting of proteasome inhibition. Major strengths include careful delineation of a mechanistic pathway to define how DBT is protective. These conclusions are well-justified.

Weaknesses:

None

---

## [Author Response]

The following is the authors’ response to the previous reviews.

**Public Reviews:**

**Reviewer #1 (Public Review):**
Summary:Through an unbiased genomewide KO screen, the authors identified loss of DBT to suppress MG132-mediated death of cultured RPE cells. Further analyses suggested that DBT reduces ubiquitinated proteins by promoting autophagy. Mechanistic studies indicated that DBT loss promotes autophagy via AMPK and its downstream ULK and mTOR signaling. Furthermore, loss of DBT suppresses polyglutamine- or TDP-43-mediated cytotoxicity and/or neurodegeneration in fly models. Finally, the authors showed that DBT proteins are increased in ALS patient tissues, compared to non-neurological controls.Strengths:The idea is novel, the evidence is convincing, and the data are clean. The findings have implications for human diseases.Weaknesses:None.

Reply: We thank the reviewer for the supportive comments.

**Reviewer #2 (Public Review):**
Summary:Hwang, Ran-Der et al utilized a CRISPR-Cas9 knockout in human retinal pigment epithelium (RPE1) cells to evaluate for suppressors of toxicity by the proteasome inhibitor MG132 and identified that knockout of dihydrolipoamide branched chain transacylase E2 (DBT) suppressed cell death. They show that DBT knockout in RPE1 cells does not alter proteasome or autophagy function at baseline. However, with MG132 treatment, they show a reduction in ubiquitinated proteins but with no change in proteasome function. Instead, they show that DBT knockout cells treated with MG132 have improved autophagy flux compared to wildtype cells treated with MG132. They show that MG132 treatment decreases ATP/ADP ratios to a greater extent in DBT knockout cells, and in accordance causes activation of AMPK. They then show downstream altered autophagy signaling in DBT knockout cells treated with MG132 compared to wild-type cells treated with MG132. Then they express the ALS mutant TDP43 M337 or expanded polyglutamine repeats to model Huntington's disease and show that knockdown of DBT improves cell survival in RPE1 cells with improved autophagic flux. They also utilize a *Drosophila* models and show that utilizing either a RNAi or CRISPR-Cas9 knockout of DBT improves eye pigment in TDP43M337V and polyglutamine repeat-expressing transgenic flies. Finally, they show evidence for increased DBT in postmortem spinal cord tissue from patients with ALS via both immunoblotting and immunofluorescence.Strengths:This is a mechanistic and well-designed paper that identifies DBT as a novel regulator of proteotoxicity via activating autophagy in the setting of proteasome inhibition. Major strengths include careful delineation of a mechanistic pathway to define how DBT is protective. These conclusions are well-justified.Weaknesses:None

Reply: We thank the reviewer for the supportive comments.

**Recommendations for the authors:**

**Reviewer #1 (Recommendations For The Authors):**
The authors have addressed my concerns. I have two more suggestions:(1) Since the authors found that MG132 inhibits autophagy, which is inconsistent with previous findings that it promotes autophagy (e.g., PMID: 26648402, 30647455, 28674081), they should discuss this discrepancy in the Discussion.

Reply: We thank the reviewer for raising this point. We agree with the reviewer that it has been well known in the literature that MG132 can lead to activation of autophagy. Indeed, we have observed in this study that MG132 itself can lead to time-dependent increases in LC3II levels in the first 8 hours of the MG132 treatment (Fig. S5B). These observations reflect the adaptive response of the cell to activate autophagy following proteasomal inhibition. However, as the MG132-mediated proteasomal inhibition persists, it is expected that the accumulation of misfolded protein substrates may overwhelm protein degradation systems, including the autophagylysosome pathway. Indeed, we have observed a reduction of the autophagic flux after 48 hours of the MG132 treatment (Fig. 3). Importantly, the DBT KO cells were able to maintain significantly higher levels of autophagic activities than the WT cells at this time point, consistent with their resistance to MG132-induced cell death. As suggested, we have added more discussion on the dynamic changes in the autophagic activities following proteasomal inhibition.

(2) A grammar issue: consider removing some of the article "the," e.g.:page 6: "the increase in cleaved PARP1 ""an increase in cleaved PARP1"; "the loss of DBT ""loss of DBT"page 7: "the loss of DBT ""loss of DBT"; "The ubiquitin modification""Ubiquitin modification"

Reply: We thank the reviewer for the supportive comments. And we have removed some of the grammar issues in the article.

**Reviewer #2 (Recommendations For The Authors):**
The authors have addressed my concerns.

Reply: We thank the reviewer for the supportive comments.